# Prenatal thalamic waves regulate cortical area size prior to sensory processing

Verónica Moreno-Juan[1],*, Anton Filipchuk[1],*, Noelia Antón-Bolaños[1],*, Cecilia Mezzera[1,2], Henrik Gezelius[1], Belen Andrés[1], Luis Rodríguez-Malmierca[1], Rafael Susín[1], Olivier Schaad[3,4], Takuji Iwasato[5,6], Roland Schüle[7,8,9], Michael Rutlin[10,11], Sacha Nelson[10], Sebastien Ducret[12], Miguel Valdeolmillos[1],*, Filippo M. Rijli[12],* & Guillermina López-Bendito[1]

The cerebral cortex is organized into specialized sensory areas, whose initial territory is determined by intracortical molecular determinants. Yet, sensory cortical area size appears to be fine tuned during development to respond to functional adaptations. Here we demonstrate the existence of a prenatal sub-cortical mechanism that regulates the cortical areas size in mice. This mechanism is mediated by spontaneous thalamic calcium waves that propagate among sensory-modality thalamic nuclei up to the cortex and that provide a means of communication among sensory systems. Wave pattern alterations in one nucleus lead to changes in the pattern of the remaining ones, triggering changes in thalamic gene expression and cortical area size. Thus, silencing calcium waves in the auditory thalamus induces *Rorβ* upregulation in a neighbouring somatosensory nucleus preluding the enlargement of the barrel-field. These findings reveal that embryonic thalamic calcium waves coordinate cortical sensory area patterning and plasticity prior to sensory information processing.

[1] Instituto de Neurociencias de Alicante, Universidad Miguel Hernández-Consejo Superior de Investigaciones Científicas (UMH-CSIC), 03550 Sant Joan d'Alacant, Spain. [2] Champalimaud Neuroscience Programme, Champalimaud Centre for the Unknown, 1400-038 Lisbon, Portugal. [3] NCCR frontiers in Genetics, University of Geneva, CH-1211 Geneva 4, Switzerland. [4] Department of Biochemistry, Sciences II, University of Geneva, CH-1211 Geneva 4, Switzerland. [5] Division of Neurogenetics, National Institute of Genetics (NIG), Mishima 411-8540, Japan. [6] Department of Genetics, SOKENDAI (The Graduate University for Advanced Studies), Mishima 411-8540, Japan. [7] Urologische Klinik und Zentrale Klinische Forschung, Klinikum der Universität Freiburg, Breisacherstrasse 66, 79106 Freiburg, Germany. [8] BIOSS Centre of Biological Signalling Studies, Albert Ludwigs University, 79106 Freiburg, Germany. [9] Deutsches Konsortium für Translationale Krebsforschung (DKTK), Standort Freiburg, 79108 Freiburg, Germany. [10] Department of Biology and National Center for Behavioral Genomics, Brandeis University, Waltham, Massachusetts 02454, USA. [11] Department of Biochemistry and Molecular Biophysics, HHMI, Columbia University Medical Center, New York, New York 10032, USA. [12] Friedrich Miescher Institute for Biomedical Research, Maulbeerstrasse 66, 4058 Basel, Switzerland. * These authors contributed equally to this work. Correspondence and requests for materials should be addressed to G.L.-B. (email: g.lbendito@umh.es).

Sensory systems are represented in the primary sensory areas of the brain in organized maps. In the embryo, these territories are pre-patterned by restricted gene expression independently of external inputs[1–5]. However, sensory cortical areas are malleable later in life as their position and size varies in function of peripheral stimuli. For example, deprivation or the loss of sensory stimuli in the visual or somatosensory systems leads to a reduction in the size of the corresponding primary cortical area and altered map representations[6–9]. Moreover, spontaneous network activity from sensory peripheral neurons also modulates the formation of cortical maps prior to sensory experience[10,11]. This is the case of retinal waves that direct map refinement in the superior colliculus and visual cortex through spatiotemporal patterns of peripheral activity[12–14]. Thus, this bottom-up plasticity, peripheral-to-central, is a well-defined mechanism that modulates cortical maps within a given sensory system.

Conversely, central structures such as the thalamus, can also influence intra-modally sensory cortical areas prior to sensory experience. Genetic manipulation of visual or somatosensory thalamocortical axons (TCAs) during embryogenesis perturbs the formation of the corresponding cortical sensory map[15,16]. Yet when a sensory input is lost early in life, thalamocortical circuits can reorganize and this change is correlated with adaptations in the size of the sensory cortical area related to the lost input[17–19]. Furthermore, top-down plasticity for the somatosensory system has also been demonstrated recently, whereby the size of the cortical barrel-field modifies its representation in subcortical sensory nuclei[20]. Thus, it is clear that both peripheral and central structures have a key role in modulating the size of cortical areas and territories within a given sensory system.

Intriguingly, the plastic changes that occur in the cortex of sensory-deprived animals involve both the deprived and spared cortical areas. For example, removal of the eyes at birth leads to a reduction of the primary visual cortex and an expansion of the somatosensory cortical barrel-field in blind adult rodents[21–23]. Thus, there would appear to be some communication among distinct sensory systems and cortical areas, although the mechanisms that underlie such effects remain unexplored.

Here we describe the existence of thalamic spontaneous calcium waves that have a specific pattern of propagation among distinct sensory-modality nuclei. We hypothesize that thalamic waves may have a pivotal role in regulating the development of cortical representations from different sensory modalities. Abolishing spontaneous calcium waves in the auditory nucleus of the thalamus alters the pattern of spontaneous waves in the neighbouring somatosensory ventral posterior medial (VPM) nucleus, and equivalent changes were also observed in embryonically enucleated mice. The increased frequency of waves in the VPM precedes an enlargement of the cortical barrel-field in S1. Mechanistically, we found activity-dependent regulation of the nuclear orphan receptor *Rorβ* in the VPM, which produced an increase in the complexity of TCA terminals. Gain- and loss-of-function experiments offer further support to the hypothesis that *Rorβ* expression in the thalamus is a key regulator of sensory cortical area adaptation.

In summary, our findings reveal a novel mode of communication between distinct sensory-modality thalamic nuclei, whereby spontaneous calcium waves control gene expression and trigger cortical size adaptations prior to the onset of sensory information processing.

## Results

**Visual embryonic deprivation expands the barrel-field.** It is well known that eye enucleation at early postnatal stages triggers a profound reorganization of deprived and non-deprived sensory cortical areas[8,24]. For instance, the somatosensory cortical area size is expanded in blind animals[8,25] while the visual primary area is reduced[8,24,26]. To determine if the development of sensory cortical territories is coordinated among distinct sensory systems already at prenatal stages, we performed bilateral eye enucleation at embryonic day (E) 14.5 (embBE), before retinal axons reach the thalamus. Experiments were done in wild-type or in a transgenic mouse line that expressed the membrane-bound enhanced green fluorescent protein (GFP) from the sensory thalamus (TCA-GFP mouse[27]). The absence of retinal axons was verified using dye tracing or a retinal ganglion cell specific transgenic mouse, $Brn3b^{Cre/+}$, crossed with a $R26^{tdTomato}$ reporter line (Fig. 1a and Supplementary Fig. 1a). We then assessed the development of the visual, somatosensory and auditory cortical areas size at early postnatal stages. Measurements of the primary cortical areas size showed a 33.3% decrease in V1 (control: $100 \pm 3.11\%$, $n = 13$; embBE: $66.7 \pm 1.88\%$, $n = 12$) and a 13.6% expansion of S1 (control: $100 \pm 1.73\%$, $n = 13$; embBE: $113.7 \pm 2.28\%$, $n = 12$; Fig. 1b,c). The A1 area did not change after embBE. The decrease in the V1 area in the embBE mice was accompanied by a similar 39.8% reduction in the dorsal lateral geniculate nucleus (dLGN) size (Supplementary Fig. 1b–d). This reduction has been reported before in early postnatal enucleated mice[26]. Interestingly, the size of the VPM did not change after embBE (Supplementary Fig. 1b–d) suggesting that the expansion of S1 might be triggered by changes in the axonal arborization of VPM thalamic neurons. To test this, we closely look at the developmental progression of the primary somatosensory cortex (S1) that displays a topographical representation of the whisker pad along the feed-forward pathway from the brainstem to the barrel field in S1 (refs 6,28,29). Immunostaining for the vesicular glutamate transporter 2 (vGlut2) that specifically labels TCA terminals in cortical layer IV, revealed a 10.7% increase in the total posteromedial barrel subfield (PMBSF) area of embBE mice at P4 (control: $100 \pm 2.63\%$, $n = 10$; embBE: $110.7 \pm 3.58\%$, $n = 14$; Fig. 1d,e) and a 12.4% increase at P8 (Supplementary Fig. 2a,b). The size of each individual barrel was also greater in the embBE mice than in their control littermates at both P4 (17.7% mean increase; control: $100 \pm 2.75\%$, $n = 10$; embBE: $117.7 \pm 5.07\%$, $n = 14$; Fig. 1f) and P8 (18.7% mean increase; Supplementary Fig. 2a,b). Moreover, these changes persisted in the adult (P30) embBE mice, where the PMBSF was increased by 11.25% as compared with control animals (Supplementary Fig. 2c,d).

Although active whisking is not performed until P10-P13 in mice[30], to unequivocally demonstrate that the barrel enlargement is independent of an increase in passive whisker flickering in young pups, we trimmed the whiskers of embBE mice daily from birth until P4. At P4, the increase in the size of the barrel area in embBE animals whose whiskers were trimmed (dewhiskered) was similar to that of embBE mice with untrimmed whiskers (control: $100 \pm 3\%$, $n = 13$; control dewhiskered P0-P4: $103 \pm 2.8\%$, $n = 16$; embBE: $113.6 \pm 3.5\%$, $n = 13$; embBE dewhiskered P0-P4: $118.6 \pm 4.2\%$, $n = 13$; Fig. 1g). Moreover, the size of barrelettes in the brainstem spinal sensory (SpV) and principal sensory (PrV) trigeminal nuclei (Supplementary Fig. 3), and of barreloids in the somatosensory VPM thalamic nucleus (Fig. 1h) was normal in early postnatal embBE animals. This strongly suggested that the enlargement of the S1 cortical area in embBE animals is not due to changes occurring at the level of either brainstem somatosensory nuclei or their presynaptic input to the thalamus. The adaptation in S1, however, might rather depend on reorganization of thalamocortical input, well before visual sensory experience.

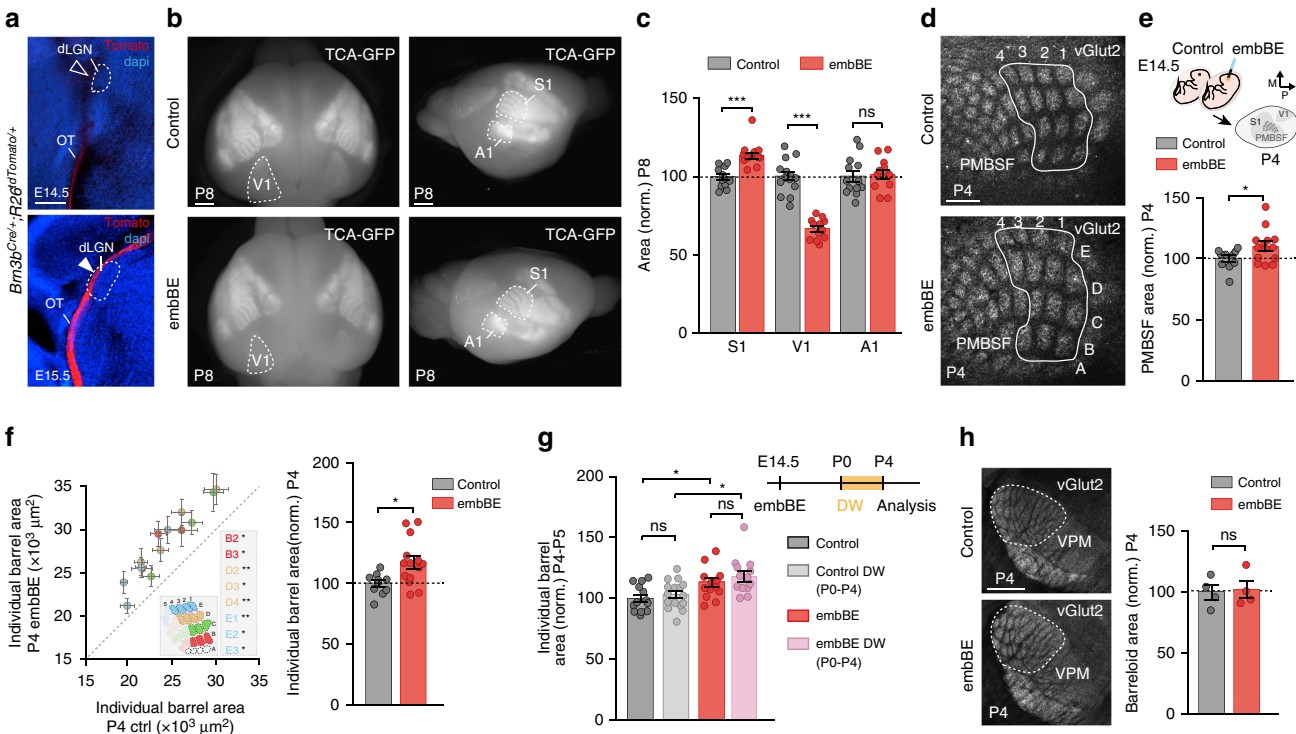

**Figure 1 | Embryonic eye removal triggers experience-independent cross-modal changes in S1 somatosensory cortex.** (**a**) $Brn3b^{Cre/+};R26^{tdTomato}$ mouse shows the absence of retinal axons in the dLGN at E14.5 and their presence at E15.5. (**b**) Labelling of principal sensory cortical areas at P8 in a control TCA-GFP transgenic mouse or in a TCA-GFP mouse in which bilateral enucleation has been performed embryonically. (**c**) Quantification of the areas of S1, V1 and A1 shown in **b** (***$P < 0.001$ for S1 and V1; not significant (ns): $P = 0.76$ for A1; Two-tailed Student's $t$-test). (**d**) vGlut2-immunostaining in the posteromedial barrel subfield (PMBSF) of S1 in control ($n = 10$) and embBE ($n = 14$) mice at P4. (**e**) Experimental design and quantification of the total PMBSF area shown in **d** (*$P = 0.03$; Two-tailed Student's $t$-test). The expansion of the PMBSF in the embBE was proportional along the medio-lateral axis (630.30 ± 39.04 pixels in control and 670.71 ± 38.79 pixels in embBE) and the anterio-posterior axis (326.60 ± 18.66 in control and 337.93 ± 26.35 in embBE mice. (**f**) Plot of the area of each individual barrel (left) and quantification of the mean individual barrel area (right) in control and embBE brains at P4 (*$P = 0.011$; Two-tailed Student's $t$-test). Inset describes the barrels that are significantly expanded in the embBE mice compared with controls. (**g**) Design of the experiment and quantification of the individual barrel area of control ($n = 13$), control dewhiskered ($n = 16$), embBE ($n = 12$) and embBE dewhiskered ($n = 13$) mice at P4 (*$P < 0.05$; ns, not significant; Two-way ANOVA test with Tukey's *post hoc* analysis). Interaction between dewhiskering and embBE was not significant ($P = 0.77$). (**h**) vGlut2-immunostaining in the VPM nucleus of the thalamus in control and embBE mice at P4, quantification of the total barreloid area (control 100 ± 6.23%, $n = 4$; embBE: 102.1 ± 7.11%, $n = 4$; $P > 0.99$; Mann–Whitney $U$-test). Graphs represent mean ± s.e.m. Scale bars, 1 mm in **b** and 300 μm in **a**,**d**,**h**.

**Thalamic calcium waves communicate sensory thalamic nuclei.**
In subcortical structures, spontaneous network activity within a given sensory modality has been shown to shape the spatial and functional organization of the corresponding sensory cortical area[12,31,32]. The thalamus is the first sub-cortical structure where the peripheral and sub-cortical inputs of visual, auditory and somatosensory circuits converge. Moreover, the results in embBE mice suggested that communication in the thalamus among distinct sensory-modality thalamocortical afferent nuclei might take place to regulate inter-areal cortical size before sensory experience. Spontaneous activity could be a promising candidate to provide such inter-nuclear communication.

We took advantage of the $Gbx2^{CreER};R26^{tdTomato}$ mice to analyse $Ca^{2+}$ signalling in thalamocortical slices in which the thalamic sensory nuclei are labelled (Fig. 2a). We found a consistent pattern of activity characterized by the emergence of thalamic $Ca^{2+}$ waves in the principal sensory nuclei (VPM, dLGN and ventro medial geniculate (MGv)) lasting from E14.5 up to P2 (Supplementary Fig. 4a–d). Thalamic waves also involve higher-order nuclei at perinatal stages. This pattern of activity was clearly distinct from the asynchronous $Ca^{2+}$ transients recorded in single cells and the synchronous activity seen in small clusters (Fig. 2b and Supplementary Fig. 4e,f). Waves propagated

with a mean front speed of 141.93 ± 29.3 μm per second, leading to a delay in the onset of the $Ca^{2+}$ transient in cells located progressively distant to the wave origin (Fig. 2c). Spatially, thalamic waves propagated across distinct sensory-modality thalamic nuclei, from one nucleus to another (VPM-dLGN, VPM-MGv and vice versa; Fig. 2d and Supplementary Movies 1–3) with a frequency of 0.20 ± 0.04 waves per minute and a duration of 7.45 ± 0.26 s (Supplementary Fig. 4g). The analysis of inter-wave intervals showed a wide distribution, ranging from 15 to more than 400 s, concentrated in the 0–3 min interval but without a defined peak of occurrence (Supplementary Fig. 4h). In order to test whether this thalamic form of activity is transmitted to the cortex, we performed calcium imaging in transgenic mice that specifically expresses the calcium indicator protein GCaMP6 in the sensory thalamic neurons (Fig. 2e). Remarkably, analysis of the activity pattern at E16.5 showed that thalamic waves are propagated through the TCAs reaching cortical areas (Fig. 2f,g; Supplementary Movie 4).

**Thalamic waves are propagated via gap junctions.** Next, we further investigated the properties and underlying mechanisms of waves initiation and propagation. At E16.5, thalamic waves were

originated from the three principal sensory thalamic nuclei with a higher frequency of origin at the VPM (Fig. 3a,b). Irrespective of their origin, the maximum area of propagation was constant for successive waves that consistently travelled across the same

territory (Fig. 3a). Within a given nucleus, the origins of waves were randomly distributed, without a regular trigger zone (Fig. 3c). To determine the possible mechanisms involved in the waves initiation, we checked the effect of the voltage-sensitive Na

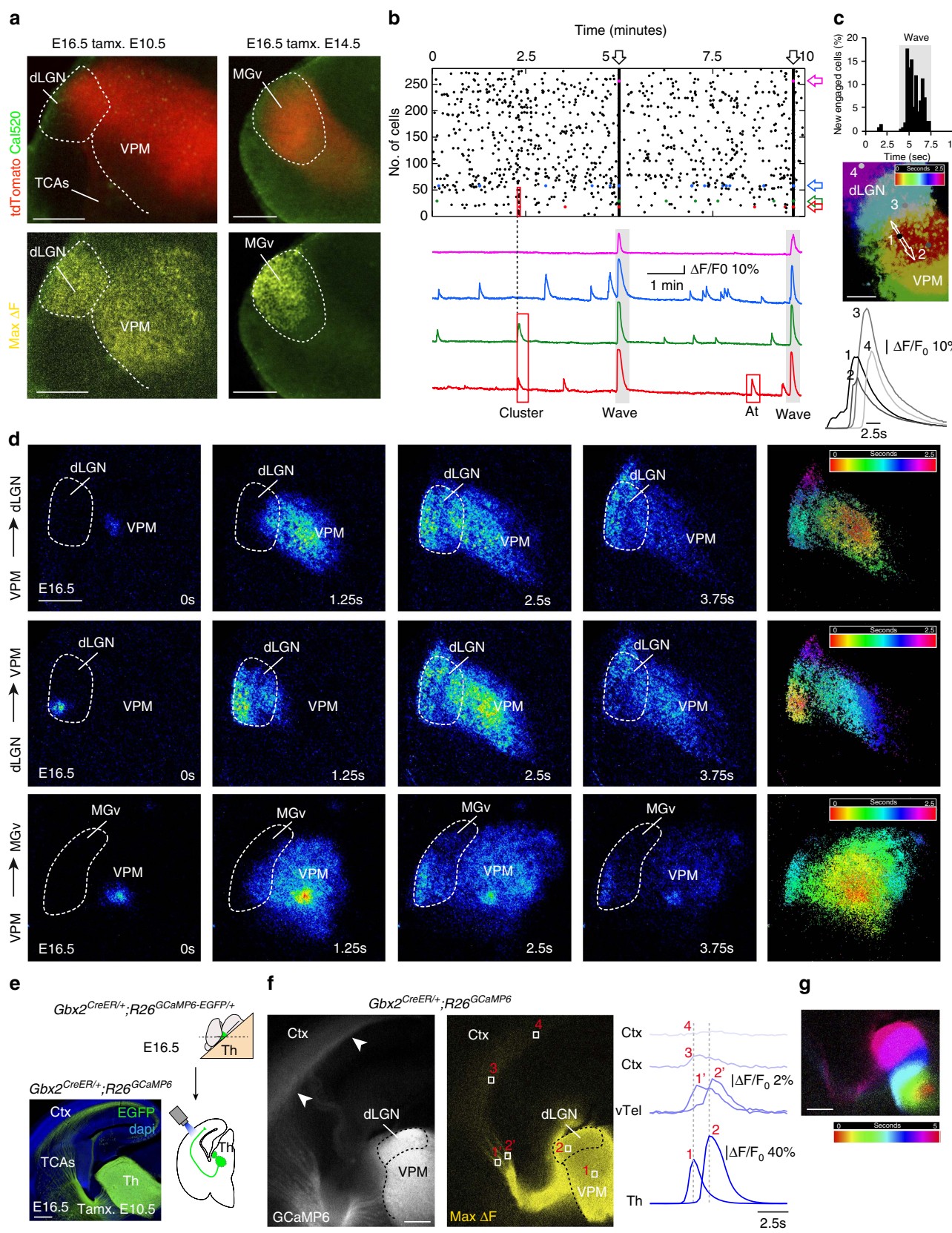

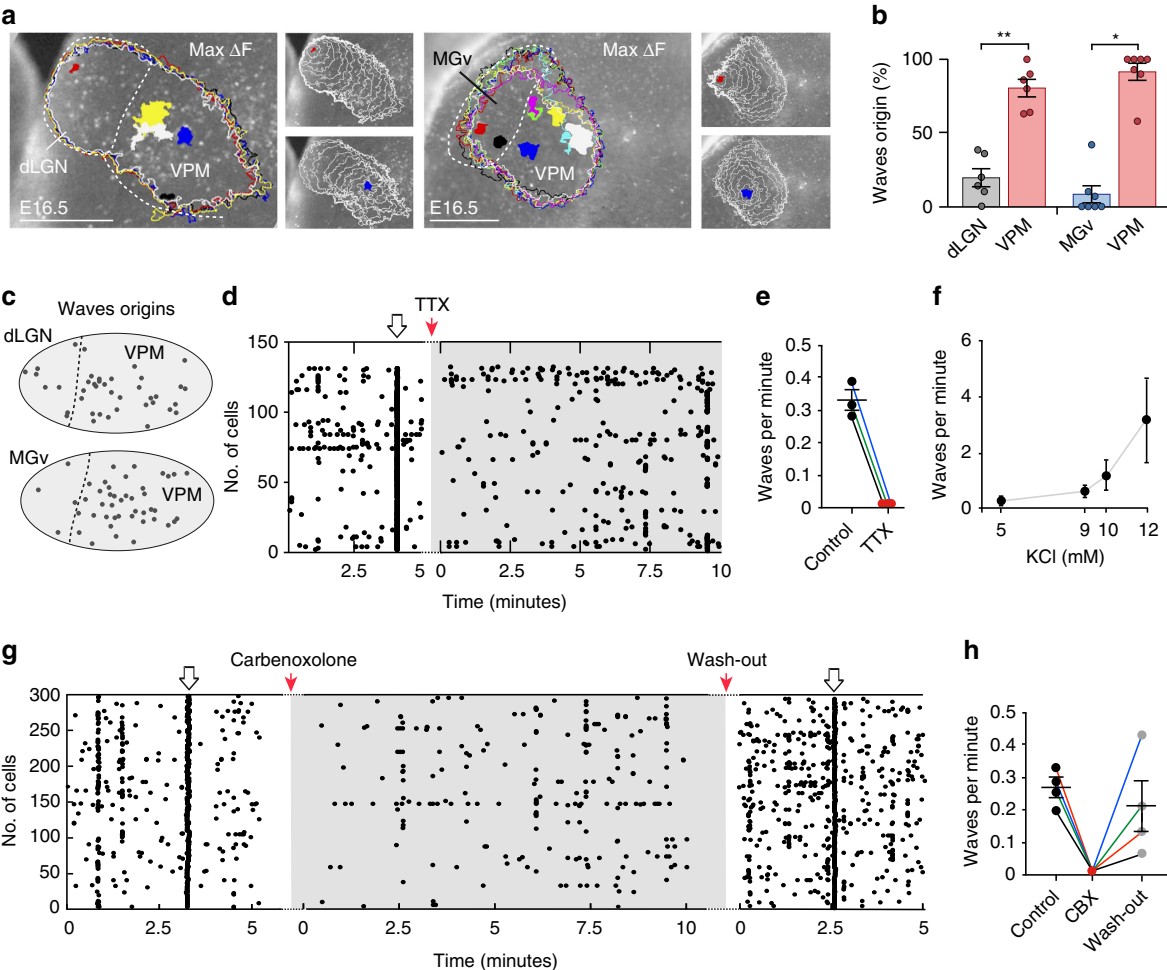

**Figure 3 | Origin and propagation properties of thalamic waves.** (**a**) Site of origin (colour-coded filled contours) and corresponding maximum spread (colour outlines) of five successive waves originated in the dLGN–VPM and eight waves in the MGv–VPM nuclei. The area covered by the waves in the dLGN–VPM example was 149.728 $\mu m^2 \pm 5.873\,\mu m^2$ and in the MGv–VPM example was 188.715 $\mu m^2 \pm 16.722\,\mu m^2$. Insets represent examples of the propagation of wave 1 (red) and wave 2 (blue) for each pair. The white contours show the pattern of spread of the wave front measured at 250 ms intervals. (**b**) Quantification of the percentage of waves depending on the origin at E16.5. The vast majority of waves are originated in the VPM (\*\*$P = 0.004$; Paired $t$-test; \*$P = 0.016$; Wilcoxon matched-pairs signed rank test). (**c**) Schemas representing the stochastic nature of the thalamic waves origins for each pair of nuclei (dLGN–VPM; $n = 40$ sites in five independent experiments; MGv–VPM, $n = 67$ sites in five independent experiments). (**d**) Effect of bath application of the voltage-dependent sodium channels blocker tetrodoxin (TTX, $1\,\mu M$) on the $Ca^{2+}$ activity. TTX completely abolished the thalamic waves without substantially affecting the asynchronous activity or the clusters. (**e**) Quantification of $Ca^{2+}$ dLGN–VPM waves frequency before and during TTX administration. (**f**) Dose-dependent response in waves per minute, after increasing the extracellular potassium concentration from 5 mM (control) to 12 mM. (**g**) Effect of bath application of the gap junction blocker carbenoxolone ($50\,\mu M$) on the $Ca^{2+}$ activity. The most noticeable effect is the reversible abolition of the synchronous waves. (**h**) Quantification of $Ca^{2+}$ dLGN–VPM waves frequency before, during and after carbenoxolone administration. Graphs represent mean ± s.e.m. Scale bars, $250\,\mu m$.

**Figure 2 | Thalamic spontaneous waves drive the communication between distinct thalamic nuclei.** (**a**) Fluorescence images of E16.5 *Gbx2*$^{CreER/+}$; *R26*$^{tdTomato}$ 45 degrees thalamocortical acute slices loaded with the calcium indicator Cal520. Areas corresponding to the dLGN, VPM and MGv nuclei express tomato in function of the time of tamoxifen administration (upper panels). Maximum projection of $Ca^{2+}$ waves (yellow) covering the three principal thalamic nuclei (lower panels). (**b**) Raster plot of the activity recorded in more than 250 individual cells in the dLGN-VPM during 10 min. The arrows label synchronous $Ca^{2+}$ transients corresponding to $Ca^{2+}$ waves. The lower panel shows examples of $Ca^{2+}$ activity traces in four individual thalamic neurons (indicated by colour arrows in the raster plot) illustrating the three patterns of $Ca^{2+}$ transients: asynchronous scattered, synchronous clusters and waves. (**c**) Cell-by-cell wave propagation. Upper panel: Percentage of neurons (over both dLGN and VPM nuclei) that are activated at every time point during wave propagation. Middle panel: Temporal spread of a wave front (colour coded). Lower panel: Examples of $Ca^{2+}$ transients in four cells during a wave indicated in the middle panel (cell 1 initiated the wave). (**d**) Propagation of thalamic waves at E16.5 in acute slices: from the VPM into the dLGN (upper panels); from the dLGN into the VPM (middle panels); and from the VPM into the MGv nuclei (lower panels). The calcium signal intensity is coded in pseudocolour. Right panels for each wave show the temporal colour coded spread of the wave front from its origin (red zone) up to the borders of the nuclei. (**e**) Expression of GCaMP6-EGFP in embryonic acute thalamocortical slice from *Gbx2*$^{CreER/+}$;*R26*$^{GCaMP6-EGFP/+}$ mice with tamoxifen administrated at E10.5. (**f**) Acute slice showing GCaMP6 in the thalamocortical projections at E16.5 (left). Maximum projection of a $Ca^{2+}$ wave from the same slice (right, yellow) showing the propagation of the thalamic waves to the cortex. Traces showing the progression of a wave from VPM (1) to dLGN (2) that propagated along the TCAs (1′ and 2′) up to distinct medio-lateral cortical territories (3 and 4). (**g**) Temporal colour-coded spread of the wave shown in **f**. Scale bars, 200 $\mu m$ in **a**, 100 $\mu m$ in **c** and 200 $\mu m$ in **d**–**g**.

channel blocker tetrodotoxin (TTX). The application of 1 μM of TTX (Fig. 3d,e) led to the abolition of the calcium waves. These results suggest that wave's initiation/propagation requires the activation of voltage-dependent Na-channels. Consistently with this, the gradual depolarization of the cells by step increases in the concentration of extracellular potassium, leads to a gradual increase in the frequency of the waves (Fig. 3f).

The function of gap-junctions has been implicated in the propagation of cortical waves in the developing neocortex[33–35]. Thus, we wondered whether gap-junctions might also participate in the mechanism of thalamic waves propagation. Notably, treatment of thalamic slices with the general gap junction blocker Carbenoxolone eliminated the propagation of the $Ca^{2+}$ waves, while preserving, although reduced, single cell and cluster activity (Fig. 3g,h). Whereas connexin-36 is expressed in developing brain areas and has been implicated in developmental processes[36], treatment with the Cx36 specific gap-junction blocker Mefloquine only decreased the frequency of the $Ca^{2+}$ waves without blocking them (Supplementary Fig. 4i), suggesting that additional connexins might be involved in this process.

**Blockage of thalamic waves affects the size of cortical areas.** We have shown that spontaneous calcium waves of activity emerge in thalamic neurons from early stages of embryonic development and propagate among distinct sensory thalamic nuclei prior to the formation of cortical maps. We next investigated whether thalamic $Ca^{2+}$ waves might coordinate the size of cortical areas during prenatal development. We generated a transgenic mouse that conditionally expressed the Kir2.1 (Kir) inward rectifier potassium channel fused to the mCherry protein (referred to as $R26^{Kir}$). When crossed to the $Gbx2^{CreER}$ line, the $R26^{Kir}$ mouse allows thalamic activity to be manipulated *in vivo*. When tamoxifen was administered at E14.5, Kir was specifically overexpressed in MGv thalamic neurons in a mosaic fashion (Supplementary Fig. 5a,b; referred to as the $MGv^{Kir}$ mouse) resulting in the blockade of the waves that originated in this structure (Fig. 4a). Individual cell activity remained in the inter-wave periods, although in fewer cells (Supplementary Fig. 5c,d). Therefore, this model provides a means to address a specific role of thalamic $Ca^{2+}$ waves in influencing cortical area size.

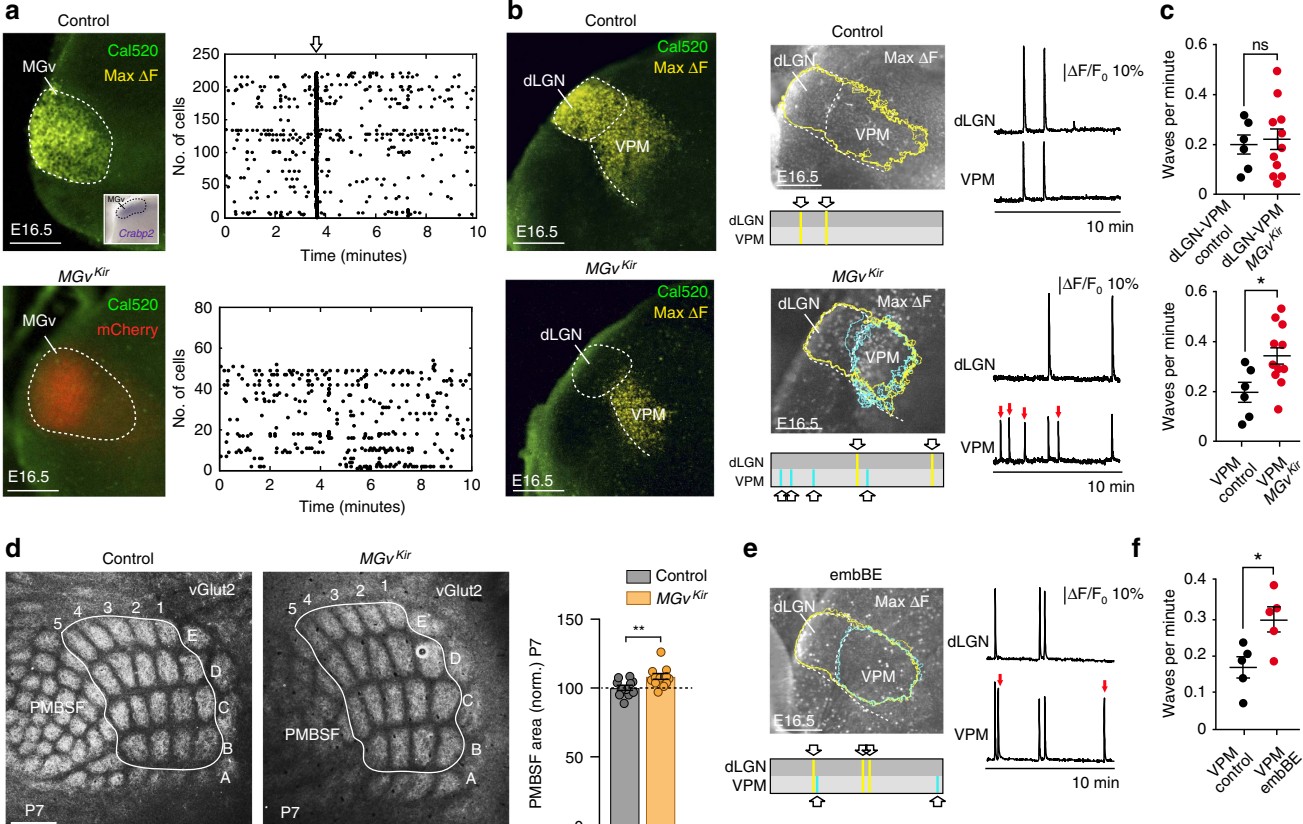

**Figure 4 | Blocking the waves in the auditory nucleus alters the pattern of wave activity in the VPM and triggers an enlargement of the barrel-field in S1.** (**a**) Maximum projection of a $Ca^{2+}$ wave (yellow) at the level of the MGv in an E16.5 control (upper panel, inset *post hoc in situ* hybridization of the auditory thalamic marker *Crabp2*) and in a $MGv^{Kir}$ (lower panel) littermate. No waves are observed in the territory were Kir2.1 is overexpressed (mCherry). Raster plot of individual cells activity recorded during 10 min in the MGv of control (upper panel) and $MGv^{Kir}$ (lower panel) mice. The $Ca^{2+}$ transient labelled by the open arrow in the control reflects a $Ca^{2+}$ wave. (**b**) Maximum projections of $Ca^{2+}$ waves (yellow) propagating between the dLGN and VPM in an E16.5 control mouse (upper panels) and in a $MGv^{Kir}$ littermate (lower panels). Additional waves (red arrows) remain restricted to the VPM in the $MGv^{Kir}$ mouse (blue contours). (**c**) Upper panel shows the quantification of $Ca^{2+}$ waves that propagate between dLGN–VPM in the control ($n = 6$) and $MGv^{Kir}$ ($n = 11$) mice showing no significant change (dLGN–VPM: 0.20 ± 0.04 waves per minute control; 0.21 ± 0.04 waves per minute $MGv^{Kir}$, $P = 0.84$). Lower panel shows the significant increase of *de novo* waves in the VPM in the $MGv^{Kir}$ mouse (VPM: 0.20 ± 0.04 waves per minute control; 0.34 ± 0.04 waves per minute $MGv^{Kir}$, *$P = 0.028$; Two-tailed Student's *t*-test). (**d**) vGlut2-immunostaining in tangential sections of the PMBSF in the S1 of P7 control ($n = 11$) and $MGv^{Kir}$ mice ($n = 12$). Quantification of the PMBSF area in P7 $MGv^{Kir}$ mice relative to the controls (**$P = 0.007$; Two-tailed Student's *t*-test). (**e**) Maximum projections of $Ca^{2+}$ waves in the embBE mouse at E17. Waves propagating between the dLGN and VPM (yellow) and *de novo* VPM waves (blue). (**f**) Quantification of the frequency of waves in the VPM in the embBE mouse (VPM: 0.16 ± 0.03 waves per minute control; 0.29 ± 0.03 waves per minute embBE, *$P = 0.019$; Two-tailed Student's *t*-test). Graphs represent mean ± s.e.m. Scale bars, 200 μm in **a,b,e,f**, and 300 μm in **d**.

The abolition of waves from the MGv resulted in an altered wave pattern in the VPM nucleus at E16.5. While the frequency of wave spreading between the dLGN and VPM nuclei remained unchanged, numerous *de novo* waves appeared in the VPM that remained restricted to this territory, significantly increasing the mean wave activity in this nucleus (control: 0.20 ± 0.04 waves per minute VPM, $n = 6$; $MGv^{Kir}$: 0.34 ± 0.04 waves per minute VPM, $n = 11$; Fig. 4b,c and Supplementary Movie 5). Despite the increase of wave frequency, the average number of calcium transients per neuron did not change in the VPM of the $MGv^{Kir}$ mouse compared with control (Supplementary Fig. 5e,f), indicating that it is the synchronization of $Ca^{2+}$ activity what was altered and not the extent of asynchronous activity. Interestingly, silencing thalamic waves in the MGv caused a 12.3% increase in the size of the S1 area (Supplementary Fig. 6a,b) with an 8% enlargement of the total PMBSF area (control: 100 ± 1.74%, $n = 11$; $MGv^{Kir}$: 108 ± 2%, $n = 12$; Fig. 4d). Individual barrels were also increased in size by 10.1% at P7 (control: 100 ± 3.07%, $n = 11$; $MGv^{Kir}$: 110.1 ± 2.85%, $n = 13$). We also found a significant decrease of 51.9% in the A1 area in the $MGv^{Kir}$ mouse (Supplementary Fig. 6a,b), suggesting that silencing thalamic waves within a thalamic sensory nucleus leads to a decrease in the size of its corresponding cortical area. No changes in the size of either barreloids in the VPM or the area of the dLGN and MGv nuclei were observed in the $MGv^{Kir}$ mice (Supplementary Fig. 6c–g).

These findings suggest that the expansion of the barrel-field in S1 might be triggered by the increased frequency of waves in VPM, rather than being a direct consequence of MGv silencing. To test the general validity of this assumption, we recorded $Ca^{2+}$ wave activity in the VPM of embBE mice at E17. As in the $MGv^{Kir}$ model, we found an increase in the frequency of waves in the VPM in the embBE mice (control: 0.16 ± 0.03 waves per minute VPM, $n = 5$; embBE: 0.29 ± 0.03 waves per minute VPM, $n = 5$; Fig. 4e,f and Supplementary Movie 6). Altogether, these results strongly suggest that prenatal thalamic waves are key regulators of sensory cortical area size and that prenatal alteration in inter-thalamic sensory nuclei communication may trigger size adaptation in cortical territories.

**Correlated activity modulates thalamic *Rorβ* gene expression.** Next, we addressed the molecular mechanism by which VPM thalamic neurons drive size adaptation of the cortical somatosensory area. In embBE mice, the enlargement of the barrel-field occurs in the first postnatal week suggesting that changes in gene expression in the perinatal VPM thalamic neurons might underlie these cortical adaptations. Hence, we compared the VPM gene expression profiles in control and embBE animals at P0 and P4 (that is, before and at the time of barrel formation) (Fig. 5a). A microarray analysis identified 106 genes, whose expression changed significantly in the VPM of embBE mice at P0 (Fig. 5b). Moreover, most of the VPM genes that changed significantly were upregulated at P0 and P4 (Fig. 5b). These results demonstrate that peripheral deprivation from a given sensory-modality (for example, visual) triggers changes in gene expression in the non-deprived thalamic somatosensory VPM nucleus. These cross-modal gene expression changes could not be explained by either TCA rewiring or major sub-thalamic reorganization of thalamic afferents (Supplementary Fig. 7).

Among the genes differentially expressed in the VPM, the RAR-related orphan receptor B (*Rorβ*)[37] was within the top ten genes significantly upregulated in the VPM (1.9-fold) in embBE mice at P0 (Fig. 5b,c). *Rorβ* was shown to be implicated in somatosensory cortical development[38] and in the postnatal cortex, *Rorβ* expression is layer-specific, with strong expression

in layer IV neurons of primary cortical areas[39]. In the thalamus, *Rorβ* is expressed in dLGN, VPM and auditory medial geniculate body neurons[39]. Between P0 and P4, *Rorβ* expression was enhanced in the VPM of control (2.4-fold) and embBE (1.6-fold) mice (Fig. 5c). *In situ* hybridization at P0 and P4 confirmed the upregulation of *Rorβ* expression in the VPM nucleus of embBE mice, mainly in its most dorsal lateral portion, the region where barreloids develop (Fig. 5d). The expression of cholecistokinine (*Cck*), a gene enriched in the VPM[40], was unaffected in the VPM of the embBE mouse at P0 (Supplementary Fig. 8a). The expression of *Rorβ* in other principal thalamic nuclei, as the MGv, did not change in the embBE mouse (Supplementary Fig. 8b). Thus, the increase of *Rorβ* expression in the somatosensory thalamic neurons predates the enlargement of the PMBSF in embBE mice.

We next tested whether spontaneous activity might regulate *Rorβ* expression in VPM, as it is shown to modulate changes in gene expression in thalamic neurons[41]. We first analysed the levels of *Rorβ* expression in VPM neurons when thalamic spontaneous activity has been artificially increased by incubation of slices with a high concentration of extracellular potassium (KCl). The increase of activity leads to a significant increase in *Rorβ* expression levels as shown by quantitative PCR (Fig. 5e) strongly suggesting an activity-dependent regulation of *Rorβ* expression. Then, we tested whether in the $MGv^{Kir}$ mouse in which there is a prenatal increase in the frequency of $Ca^{2+}$ waves in the VPM, without overall increase of $Ca^{2+}$ spontaneous activity (Supplementary Fig. 5f), the *Rorβ* expression levels are also upregulated. Indeed, analysis of *Rorβ* expression in the $MGv^{Kir}$ mouse demonstrated the upregulation of messenger RNA levels in VPM neurons both by quantitative PCR at E16.5 and by *in situ* hybridization at P0 (Fig. 5f,g). Thus, prenatal *Rorβ* expression in VPM neurons might be regulated by synchronous spontaneous thalamic activity. Moreover, we found that the expression of *Rorβ* in the MGv of the $MGv^{Kir}$ mouse is decreased at P0 (Supplementary Fig. 8c,d), which is correlated with the decrease of A1 size. Altogether, these results indicate that *Rorβ* expression is positively regulated by correlated spontaneous activity.

**Rorβ enhances TCA branching and controls barrel-field size.** The fact that *Rorβ* is expressed more strongly in VPM neurons after enucleation and MGv silencing, prompted us to hypothesize that thalamic *Rorβ* might be an important element in the control of the somatosensory system developmental program. To address the role of *Rorβ*, we first examined the size of the barrel-field territory in a conditional $Nestin^{Cre/+};Rorβ^{fl/fl}$ mice at P8 (Fig. 6a), and found that in the absence of *Rorβ* the total PMBSF area was 18.1% smaller ($Nestin^{+/+};Rorβ^{fl/fl}$: 100 ± 3.33%, $n = 7$; $Nestin^{Cre/+};Rorβ^{fl/fl}$: 81.9 ± 2.93%, $n = 5$; Fig. 6b) compared with their control littermates. Furthermore, a 21.8% decrease in the size of individual barrels was also evident at this stage at P8 ($Nestin^{+/+};Rorβ^{fl/fl}$: 100 ± 3.49%, $n = 8$; $Nestin^{Cre/+};Rorβ^{fl/fl}$: 78.2% ± 2.19%, $n = 7$; Fig. 6c). A similar reduction was observed in $Rorβ^{-/-}$ mice at P4 (Supplementary Fig. 9).

We next examined the effect of *Rorβ* gain of function on thalamic axon growth and branching in dissociated thalamic neurons *in vitro*. *Rorβ* overexpression in thalamic neurons provoked an increase in axon complexity and neurite length (*i-Gfp*: 880.3 ± 68.9 μm, $n = 33$ neurons from three independent experiments; *Rorβ-i-Gfp*: 1,727 ± 130.7 μm, $n = 29$ neurons from three independent experiments), as compared with control neurons (Fig. 6d,e). To assess whether the size of the PMBSF area might be controlled by thalamic *Rorβ* expression, we induced *Rorβ* overexpression *in vivo* by *in utero* electroporation of VPM

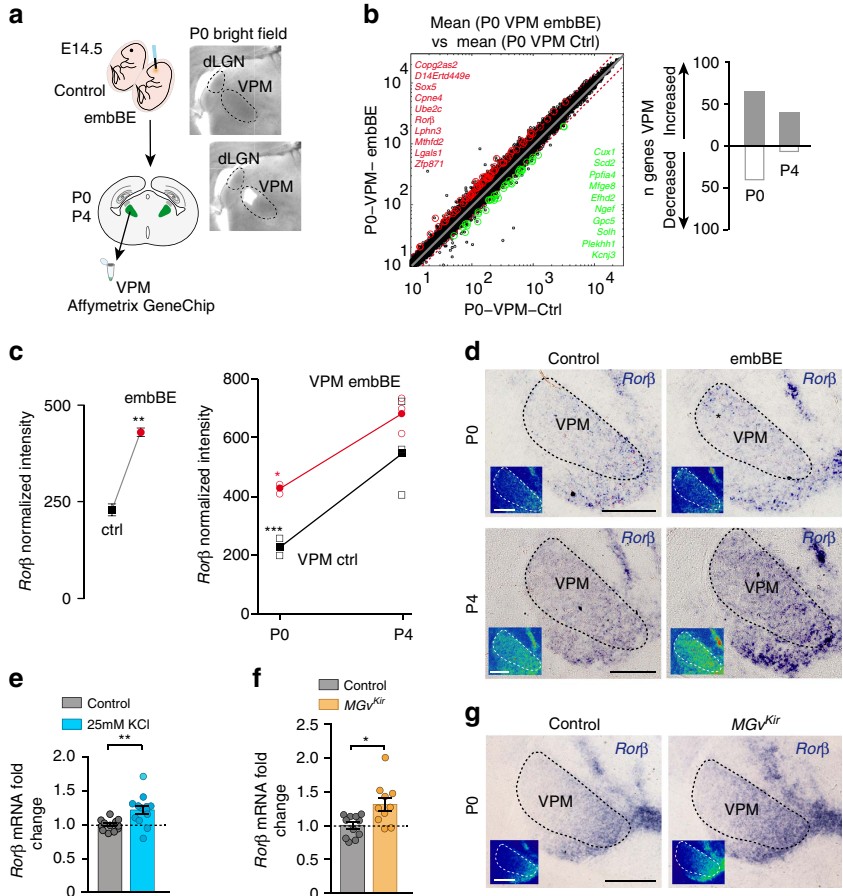

**Figure 5 | Both embBE and abolition of MGv waves induce changes in gene expression in the VPM thalamic nucleus.** (**a**) Scheme representing the microarray experiment. VPM nuclei were collected at P0 and P4, and the RNA was extracted and processed according to the Affymetrix GeneChip protocol. Bright field image showing a coronal slice after dissection of the VPM nucleus. (**b**) Scatter-plots showing significantly upregulated or downregulated transcripts (red and green circles, respectively) with a change of ≥1.5 or ≤ −1.5-fold and a *P* value of ≤0.05 at P0 and P4. In the VPM, 106 transcripts were significantly regulated at P0 and 47 at P4, with an overall tendency towards upregulation. The top ten significantly upregulated or downregulated transcripts in the VPM are listed in red or green, respectively. (**c**) The expression of the *Rorβ* gene in VPM nucleus is upregulated by 1.9-fold at P0 in embBE mice (\*\**P* = 0.008; Two-tailed Student's *t*-test). Between P0 and P4, *Rorβ* expression is upregulated in the VPM 2.4-fold in control mice (\*\*\**P* < 0.001; Two-tailed Student's *t*-test) and 1.6-fold in embBE mice (\**P* = 0.04; Two-tailed Student's *t*-test). (**d**) *In situ* hybridization for *Rorβ* in coronal sections from control and embBE animals at P0 and P4. Note the stronger expression of *Rorβ* in the VPM area (asterisk) of embBE animals compared with the controls. (**e**) Quantitative real-time PCR for *Rorβ* transcripts in VPM neurons, in control media (*n* = 12) or after treatment with 25 mM of KCl (*n* = 13) in acute slices at E16.5 (\*\**P* = 0.0042, Two-tailed Student's *t*-test). (**f**) Quantitative real-time PCR for *Rorβ* in the VPM nucleus in control (*n* = 11) and *MGv^Kir^* (*n* = 10) mice at E16.5 (\**P* = 0.01, Mann–Whitney *U*-test). (**g**) *In situ* hybridization for *Rorβ* in coronal sections from control (*n* = 5) and *MGv^Kir^* (*n* = 5) animals at P0. Graphs represent mean ± s.e.m. Scale bars, 300 μm.

neurons at E11.5 (Fig. 6f and Supplementary Fig. 10). First we analysed the size of *Rorβ*-overexpressing TCA terminals defining individual barrels and found that the individual barrel size was 26.8% larger in *Rorβ*-overexpressing hemispheres as compared with *i-Gfp*-electroporated control (*i-Gfp*: 100 ± 4.4%, *n* = 4; *Rorβ-i-Gfp*: 126.8 ± 5.4%, *n* = 5; Fig. 6g). Moreover, single axon reconstructions of VPM neurons showed that overexpression of *Rorβ* leads to an increased axonal terminal length (*i-Gfp*: 453.6 ± 53.55 μm, *n* = 12; *Rorβ-i-Gfp*: 727.8 ± 62.96 μm, *n* = 11; Fig. 6h,i) and a larger axonal terminal area in S1 (*i-Gfp*: 100 ± 14.57%, *n* = 12; *Rorβ-i-Gfp*: 193 ± 29.3%, *n* = 12; Fig. 6h,i). Thus, expression of *Rorβ* in VPM neurons influences the total PMBSF area by modifying TCA branching *in vivo*.

## Discussion

Elucidating the mechanisms that control the developmental programs of the distinct sensory cortical areas is critical to understand area-specific sensory circuits and function. We report here a previously unrecognized prenatal role for the thalamus, which depends on propagation of spontaneous calcium waves among distinct sensory-modality nuclei (Fig. 7). These thalamic waves allow developing sensory nuclei to interact and coordinate input-specific gene expression, in turn influencing the size of primary cortical areas during development prior to sensory experience. In addition, modulating the frequency of these waves provides a means to induce adaptations in cortical territories, as seen by the increased barrel area size in embBE and *MGv^Kir^* mice.

To define the sub-cortical mechanisms involved in the prenatal control of cortical area size, we used embBE in which the visual cortical area does not receive any input from its respective sensory modality, the eyes being removed before thalamocortical afferents reach the cortex[42,43]. We found that the expansion of the PMBSF area and individual barrels in the adult embBE mouse is set during the first postnatal week, before the animal

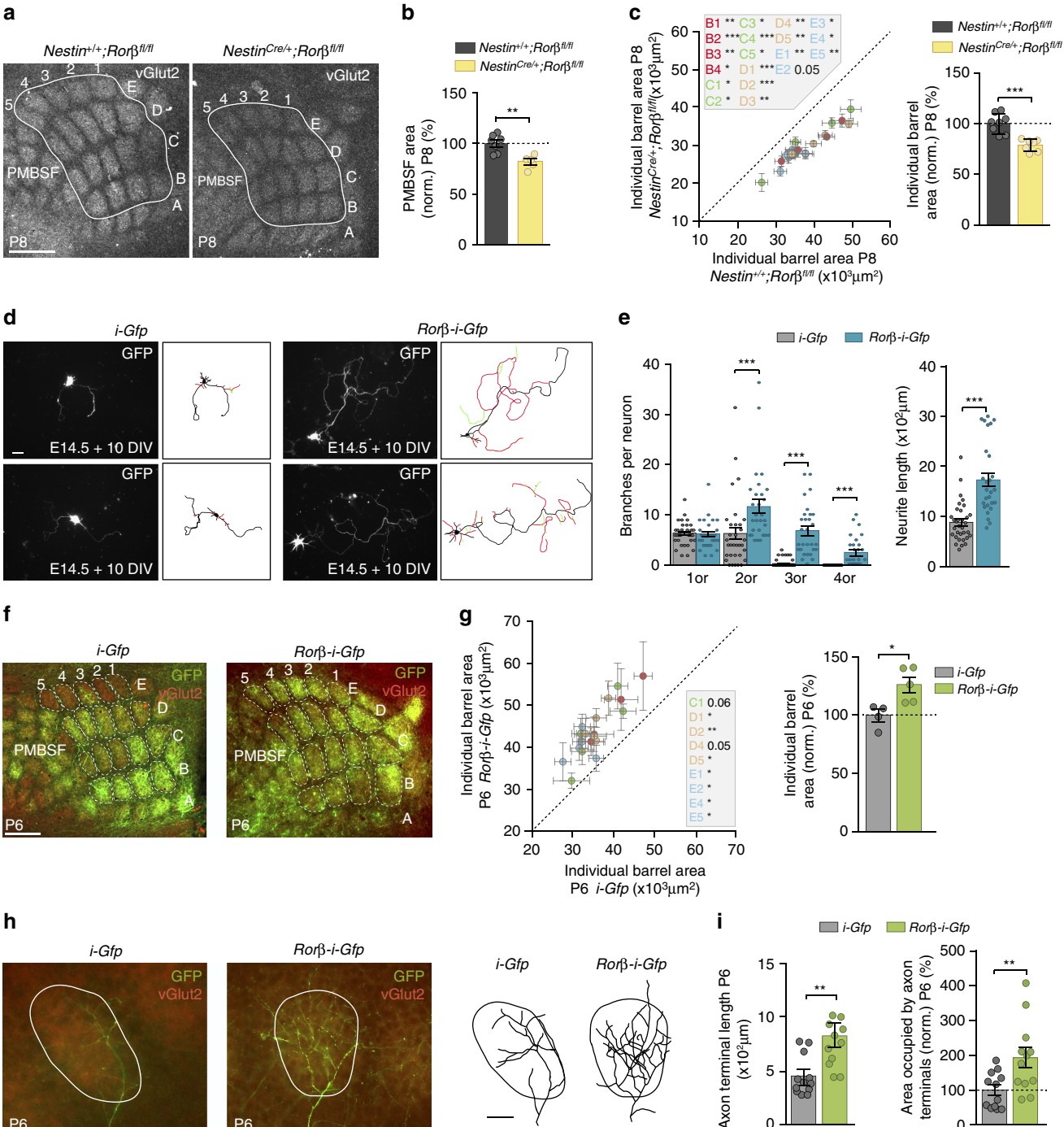

**Figure 6 | Thalamic *Rorβ* modulates the development of the somatosensory system and axonal branching.** (**a**) vGlut2-immunostaining in tangential sections of the PMBSF from control *Nestin*$^{+/+}$*;Rorβ*$^{fl/fl}$ ($n=8$) and *Nestin*$^{Cre/+}$*;Rorβ*$^{fl/fl}$ ($n=7$) mice at P8. (**b**) Quantification of the total PMBSF area shown in **a** (**$P=0.0031$; Two-tailed Student's *t*-test). (**c**) Plot of each individual barrel area and quantification of individual barrel area in control *Nestin*$^{+/+}$*;Rorβ*$^{fl/fl}$ ($n=8$) and *Nestin*$^{Cre/+}$*;Rorβ*$^{fl/fl}$ ($n=7$) brains (***$P<0.001$; Two-tailed Student's *t*-test). Insert describes the barrels that are significantly reduced in the double mutant mice. (**d**) Fluorescence images and representative drawings of thalamic neurons from E14.5 mice transfected with *i-Gfp* ($n=33$ neurons from three independent cultures) or *Rorβ-i-Gfp* ($n=29$ neurons from three independent cultures) and analysed at 10 days *in vitro* (DIV). (**e**) Quantification of branches per neuron at increasing branch orders (***$P<0.001$ for second, third and fourth branch orders; Mann–Whitney *U*-test). Quantification of the total neurite length per neuron (***$P<0.001$; Mann–Whitney *U*-test). (**f**) Flattened tangential sections showing vGlut2-immunostaining in S1 at P6 after *Rorβ-i-Gfp* ($n=5$) electroporation compared with control *i-Gfp* electroporated brains ($n=4$) at E11.5. Thalamic *Rorβ* overexpression induces the increase on the size of the individual barrel area. (**g**) Quantification of the individual barrel area at P6 in *i-Gfp*-electroporated and *Rorβ-i-Gfp* electroporated brains (*$P=0.02$; Mann–Whitney *U*-test). Insert describes the barrels that are significantly expanded after *Rorβ-i-Gfp* electroporation. (**h**) Coronal sections showing the axonal arborization of individual VPM neurons in a single barrel (immunolabelled with vGlut2) after electroporation with *i-Gfp* ($n=12$ neurons from four brains) or *Rorβ-i-Gfp* ($n=12$ neurons from nine brains). Right panel: example of single axons reconstruction under the two conditions. (**i**) Quantification of the axon terminal length and the area occupied by the axon terminals shown in **h** (**$P=0.003$ and **$P=0.009$; Two-tailed Student's *t*-test). Graphs represent mean ± s.e.m. Scale bars, 300 μm for **a** and **f**, 100 μm for **d** and 20 μm for **h**.

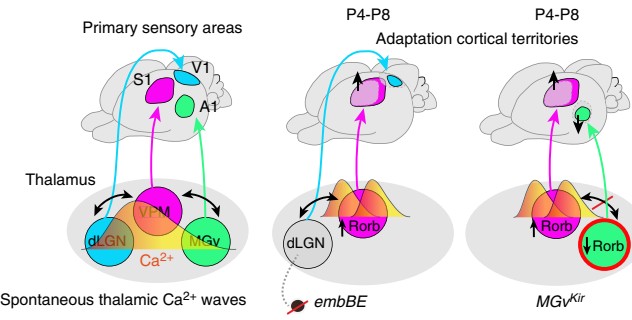

**Figure 7 | Thalamic mechanism of coordinating sensory cortical areas territories mediated by the existence of spontaneous calcium waves.** Embryonically visual input deprived mice (embBE) show an expansion of the primary somatosensory cortex (S1) prior to sensory experience. This expansion of the barrel-field is triggered by activity-dependent gene regulation in the VPM. Both the embryonic abolishment on peripheral input or silencing thalamic waves in the auditory nucleus of the thalamus (MGv) leads to an increase wave activity in the VPM, which triggers *Rorβ* expression and an expansion of the barrel-field in S1. When the MGv auditory thalamic waves are silenced, the expression of *Rorβ* is decreased and this effect predates the reduction of the A1 area.

experiences any sensory input. These results are in line with recent observations in rodents enucleated at birth, where changes in gene expression and in the size of cortical areas were detected postnatally[23,44]. The fact that the enlargement of S1 and the reduction of A1 described in the $MGv^{Kir}$ mice are not correlated with a modification in the size of the respective thalamic nuclei supports a thalamo-cortical rather than a top-down[45] mechanism for plasticity. However, whether a cortico-cortical mechanism might also contribute to the cross-modal experience independent phenomenon described here remains to be tested.

Our finding that thalamic waves emerge before peripheral input reaches the thalamus (see also ref. 46) indicates that they might be intrinsically generated in this structure. Indeed, spontaneous calcium waves persist in the absence of retinal input, although their pattern and frequency is altered if no axons from this organ are received. In sensory systems, spontaneous peripheral activity appears before the onset of natural sensory transduction and it is thought to have a key role in pattern formation[13,32]. Thus, it is likely that spontaneous activity generated peripherally will eventually override intrinsically generated thalamic waves, as suggested by the correlated activity of successive visual relay stations recorded at early postnatal stages[47,48]. It has been shown that silencing peripheral input postnatally[49], or in adulthood[50], does not eliminate synchronous thalamic sensory activity, suggesting that intrinsic and peripherally transmitted activity can co-exist.

Early neuronal activity can have a merely permissive role in neural circuit formation or an instructive role through specific spatiotemporal patterns of neuronal activity[51,52]. Our results show a correlation between the increase of the frequency of the waves in the VPM, in both embBE and the $MGv^{Kir}$ mice, and the enlargement of the PMBSF size. This increase in wave activity is not accompanied by a change in the total activity in individual cells in the VPM, suggesting that the patterned thalamic waves are instructive for barrel field development. Moreover, the abolishment of the calcium waves specifically in the MGv triggered a decrease of the A1 area size, again without changing the overall single cell activity, and thus supporting an instructive role of the thalamic waves in the developmental program of cortical areas.

Waves of calcium activity have been recorded in thalamic astrocytes at early postnatal stages in rat[53]. However, it is unlikely

that astrocytes contribute to the prenatal waves described here, as astrogenesis in the rodent brain takes place postnatally[54]. Spontaneous thalamic waves precisely delineate thalamic nuclei, suggesting that this activity may have a role in the early definition of these territories. Moreover, we found that waves propagate from one sensory-modality thalamic nucleus to another, with robust specificity, and that they engage a large proportion of the neurons in each structure. Hence, asynchronous activity generated by individual neurons and then amplified by connectivity through gap junctions, may recruit sufficient number of thalamic neurons to periodically initiate synchronous waves. The results from the $MGv^{Kir}$ mouse seem to support this notion, given that conditional overexpression of *Kir2.1* in the MGv provokes the disappearance of thalamic waves. However, it remains to be tested whether sensory-modality thalamic nuclei might have a specific connexins expression patterns or stronger gap junction connectivity than adjacent thalamic regions, preventing the $Ca^{2+}$ waves from surpassing their borders. Our results with TTX and Carbenoxolone that block thalamic calcium waves, suggest similar mechanisms of generation and propagation of thalamic and neocortical spontaneous synchronous activity[33,34].

Selective silencing of the waves in the auditory nucleus produces changes in the frequency of thalamic waves and in the expression of *Rorβ* in the neighbouring VPM nucleus (without affecting its size), eventually provoking an enlargement of the PMBSF. Our data suggest that prenatal thalamic *Rorβ* expression is a regulator of the size of the barrel cortex representation and they show that in the VPM, *Rorβ* is upregulated embryonically in a spontaneous activity-dependent manner. Increasing the frequency of calcium waves in the VPM, either by embryonic eye enucleation or by MGv wave silencing in $MGv^{Kir}$ mice, enhances *Rorβ* expression in the somatosensory thalamic nucleus, an event that precedes the enlargement of the PMBSF area in the S1 region. Conversely, *Rorβ* expression is downregulated in the MGv nucleus of $MGv^{Kir}$ mice where A1 area is significantly decreased, implying a more general role of *Rorβ* in modifying cortical areas size across sensory modalities.

Our data shows that the enlargements in the S1 area size shown in the embBE and $MGv^{Kir}$ mice are accompanied by an expansion of individual barrels, and thus they are probably caused by an increase in the complexity of the TCA branches that form the barrels. Indeed, increasing *Rorβ* levels in dissociated thalamic neurons *in vitro* or in the VPM neurons *in vivo*, lead to an increased length and complexity of TCAs. Altogether these results suggest a mechanism in which an activity-dependent regulation of *Rorβ* might control the expansion and plasticity of cortical areas by modulating TCA branching. Interestingly, an analysis of the *Rorβ* promoter identified several motifs that are putative targets for $Ca^{2+}$ sensitive transcription factors like CREB, AP-1 and NF-κB-binding sites[41,55,56]. In addition to the effects on *Rorβ* expression, calcium waves are transmitted to the thalamocortical terminals providing a patterned form of activity that may also contribute to shape different sensory cortical territories.

In conclusion, our results highlight how spontaneous activity-dependent mechanisms have a prominent role in the early stages of development[41,57]. We show that highly synchronized spontaneous activity in the form of $Ca^{2+}$ waves provides a means of communication among distinct sensory systems that regulates gene expression programs and that drives cross-modal adaptation of cortical sensory area size. Moreover, such a network of communication establishes the thalamus as an important sub-cortical hub of coordination between prenatal sensory systems. This information provides novel clues to understand the compensatory cortical expansion and increased capabilities observed in congenitally blind and deaf humans[58–60].

## Methods

**Mouse strains.** Wild-type mice maintained on a C57BL/6 background were used for the microarray expression profiling, whereas wild-type mice maintained on an ICR/CD-1 background were used for the *in vivo* experiments, tracing studies, $Ca^{2+}$ imaging, organotypic and primary neuronal cultures. The day on which the vaginal plug was detected was designated as E0.5. The $R26^{tdTomato}$ Cre-dependent mouse line[61] was obtained from Jackson Laboratories (Stock number 007908). The $R26^{GCaMP6f}$ Cre-dependent mouse line was obtained from Jackson Laboratories (Stock number 024105). TCA-GFP Tg line has been described previously[27]. The $R26^{Kir2.1-mCherry}$ mouse line was generated by inserting a *CAG-lox-STOP-lox-Kir2.1-mCherry-WPRE-pA* cassette into the *Rosa26* gene locus as described below. Each of the $R26$ reporter mice carry a *Rosa26* locus with a floxed STOP cassette that prevents the expression of the gene indicated. The reporter mice were crossed with an inducible thalamic-specific $Gbx2^{CreER}$ line[62] in order to generate $Gbx2^{CreER/+};R26^{X/+}$ double mutant embryos. The $Gbx2^{CreER}$ line expresses *CreER(T2)-ires-eGfp* under the control of the Gbx2 promoter. Tamoxifen induction of Cre recombination in the double mutant embryos was performed by gavage administration of tamoxifen (5–7 mg dissolved in corn oil, Sigma) at E14.5 to specifically target the MGv thalamic nucleus or at E10.5 to label all principal thalamic nuclei. The $R26^{tdTomato}$ were crossed with the retinal ganglion cell specific $Brn3b^{Cre}$ line (courtesy of Dr. Vann Bennett) in order to generate $Brn3b^{Cre/+};R26^{tdTomato/+}$ double mutant embryos. The $Ror\beta^{-/-}$ mouse has been described previously[63]. The $Ror\beta^{fl}$ mice were crossed to a $Nestin^{Cre}$ line previously described[64] (courtesy of Angel Barco; Jackson 003771) to conditional delete $Ror\beta$ from all neurons. All the transgenic animals used in this study were maintained on an ICR/CD-1 or C57BL/6 genetic background and all the animals were genotyped by PCR. The Committee on Animal Research at the University Miguel Hernández approved all the animal procedures, which were carried out in compliance with Spanish and European Union regulations.

**Generation of the $R26^{Kir2.1-mCherry}$ mouse line.** The human *Kir2.1* gene was fused to *mCherry* by removing the stop codon and inserting mCherry in frame with a NotI linker using TOPO (Invitrogen) and conventional cloning. The resulting *Kir2.1-mCherry* fusion gene was then introduced into a *Rosa26-CAG-lox-STOP-lox* targeting construct (Ai27, a gift from Hongkui Zeng, Addgene plasmid #34630 (refs 61,65), replacing the existing insert by conventional cloning. The resulting targeting construct was electroporated into E14 ES (129Sv Ola) cells and colonies with successful insertion were aggregated with morula-stage embryos obtained from inbred (C57BL/6 × DBA/2) F1 mice to generate chimeric mice by standard procedures. Chimeric mice were bred with C57BL/6J mice to obtain germline transmission to F1 mice. These mice were then screened for reporter expression by crossing with a Cre mouse. Two animals were selected from the same ES clone that successfully passed the construct through the germline and that displayed the expected Cre-activated expression. In the studies presented here, this line was outbred onto an ICR background to strengthen its resistance to tamoxifen treatments, as described above.

**Histology.** For *in situ* hybridization and immunohistochemistry at postnatal stages, mice were perfused with 4% paraformaldehyde (PFA) in PBS (0.01 M), and their brains were removed and post-fixed in the same fixative overnight. For immunohistochemistry of the embryonic tissue, the brains were dissected out and fixed immediately in 4% PFA overnight. Cytochrome oxidase staining was performed to label the somatosensory pathway. Cortical hemispheres were cryoprotected with sucrose and cut tangentially at 80–100 μm with a cryotome (MICRON). Sections were incubated overnight at 37 °C in a cytochrome oxidase solution (0.03% cytochrome c (Sigma), 0.05% 3,3′-diaminobenzidine (Sigma) and 4% sucrose in 0.01 M PBS). Immunohistochemistry was performed on vibratome or cryotome brain sections that were first incubated for 1 h at room temperature in a blocking solution containing 1% BSA (Sigma) and 0.3% Triton X-100 (Sigma) in PBS 0.01M. Subsequently, the sections were incubated overnight at 4 °C with the primary antibodies: rabbit anti-vGlut2 (1:500, Synaptic Systems, #135402), chicken anti-GFP (1:3,000; Aves Labs, #GFP-1020) and rat anti-RFP (1:1,000 Chromotek, #5F8). The sections were then rinsed three times in PBS 0.01 M and incubated for 2 h at room temperature with secondary antibodies: Alexa546 donkey anti-rabbit (1:500, ThermoFisher, #A10040), Alexa488 goat anti-chicken (1:500, Thermo-Fisher, #A11039), Alexa594 donkey anti-rat (1:500, ThermoFisher, #A21209), Alexa488 donkey anti-rabbit (1:500, ThermoFisher, #A21206). Finally, the sections were counterstained with the fluorescent nuclear dye Dapi (Sigma-Aldrich).

*In situ* hybridization was performed on 60–100 μm vibratome sections using digoxigenin-labelled antisense probe for *Rorβ*, *Crabp2* and *Cck*. Hybridization was carried out overnight at 65 °C, and after hybridization the sections were washed and incubated overnight at 4 °C with an alkaline phosphatase-conjugated anti-digoxigenin antibody (1/2500-1/4000, Roche). To visualize the RNA-probe binding, colorimetric reaction was performed for 1-2 days at room temperature in a solution containing NBT (nitro-blue tetrazolium chloride, Roche) and BCIP (5-bromo-4-chloro-3′-indoly phosphate p-toluidine salt, Roche). After development, the sections were washed and mounted in Glycerol Jelly (Merck Millipore).

**Dye tracing studies.** For axonal tracing at postnatal stages, animals were perfused with 4% PFA in PBS 0.01 M, and their brain was dissected out and post-fixed overnight in the same fixative. Small DiI (1,1′-dioctadecyl 3,3,3′,3′-tetramethylindocarbocyanine perchlorate; Invitrogen) or DiA (4-[4-(dihexadecylamino) styryl]-N-methylpyridinium iodide) crystals were inserted into the distinct primary cortical areas, thalamus, eyes or eye cavities, the trigeminal nucleus and the inferior colliculus. The dyes were allowed to diffuse at 37 °C in PFA solution for 1–4 weeks. Vibratome sections (60–100 μm) were then counterstained with the fluorescent nuclear dye Dapi (Sigma-Aldrich).

**In utero electroporation and in utero enucleation.** For *in utero* electroporation, pregnant females (E11.5) were deeply anaesthetized with isoflurane to perform laparotomies. The embryos were exposed and the third ventricles of the embryonic brains were visualized through the uterus with an optic fibre light source. The full-length *Rorβ* (a generous gift from Jeffrey Macklis) or a backbone construct were concentrated to 1.5 μg μl$^{-1}$ and mixed together with a plasmid encoding *Gfp* at 0.9 μg μl$^{-1}$ and 1% Fast Green (Sigma). The plasmids were injected into the third cerebral ventricle of each embryo with an injector (Nanoliter 2010, WPI). For electroporation, the negative and positive palettes were placed near the head of the embryo, and 5 square electric pulses of 45 V and 50 ms were delivered through the uterus at 950 ms intervals using a square pulse electroporator (CUY21 Edit: NepaGene Co., Japan). The surgical incision was then closed and embryos were allowed to develop until either E18.5 or P6. To analyse the brain at E18.5 it was removed and fixed directly in 4% PFA, whereas P6 mice were first perfused with 4% PFA and then processed for further analysis.

For *in utero* enucleation, the same surgical procedure was carried out on pregnant females at E14.5 but once the uterus was exposed, both eyes were cauterized in half of the embryos of each litter. The surgical incision was closed and embryos were allowed to develop until postnatal stages.

**Postnatal whisker trimming.** To perform postnatal whisker trimming in enucleated and control mice, all the whiskers were cut to the level of the guard hairs on both sides of the face daily from P0 to P4. Pups were perfused with 4% PFA, and their brain was dissected out and post-fixed overnight in the same fixative.

**Measurement of brain areas and data analysis.** ImageJ software was used to measure the size of the thalamic nuclei, the individual barrels and the PMBSF areas. For PMBSF and individual barrel areas data was normalized. In the case of the individual barrels, each barrel area from a given experimental condition was normalized to the corresponding barrel mean area in the control, which was considered as 1. To ensure consistent analysis, we choose rows B1-3, C1-3, D1-4 and E1-4 for mice analysed at P4, while B1-4, C1-5, D1-5 and E1-5 were chosen to analyse older mice. These barrels were constantly present in the slices obtained after processing the brains. For the quantification of the individual barrel area in the *in utero Rorβ* electroporations, brains with target electroporation in the VPM nucleus were selected. The electroporated side was coronally cut for the thalamus and tangentially cut for analysing the cortex. A TCA-GFP mouse was used to measure the size of the primary cortical areas and the area and volume of the distinct thalamic nuclei. In order to quantify the size of cortical areas, TCA-GFP (control, embBE and $MGv^{Kir}$) mice were perfused and directly process to obtain images under the stereo fluorescent microscope (Leica MZ10 F). Coronal serial slices of 80–100 μm were obtained from TCA-GFP brains and distinct thalamic nuclei were immunolabel with Gfp and Vglut2 in order to better detect the structures. Neurolucida explorer from MBF Bioscience was used to quantify the volume of dLGN, VPM and MGv thalamic nuclei.

**Organotypic thalamic cultures.** To analyse *Rorβ* expression after KCl treatment *ex vivo*, pregnant mice were killed and their embryos were recovered at E16.5. Organotypic slice cultures of embryonic thalamus were prepared as previously described[66]. Thalamic coronal slices were place for 5 h at 37 °C with 5% $CO_2$ in maintenance medium (Glutamax 1 ×, 45% glucose solution 50 mM, penicillin/streptomycin 100 U per microlitre, 2% B27 and 95% Neurobasal). For the experimental condition 20 mM of KCl was added. After incubation, and in order to perform RNA extraction and quantitative PCR, VPM nuclei were dissected out and collected in pulls of three embryos.

**Microdissection of thalamic nuclei.** To collect tissue from the VPM nucleus at neonatal stages, animals were killed and their brain was dissected out in RNase-free conditions to prevent RNA degradation. Vibratome sections (200 μm) were obtained and collected in ice-cold oxygenated aCSF (117 mM NaCl, 4.7 mM KCl, 1.2 mM $MgCl_2$, 2.5 mM $CaCl_2$, 1.2 mM $NaH_2PO_4$, 25 mM $NaHCO_3$ and 0.45% D-glucose), and the thalamic nucleus was rapidly microdissected under a microscope. The tissue was maintained overnight at 4 °C in RNA-Later (Sigma) and stored at −80 °C for subsequent RNA extraction.

**Affymetrix microarray.** For microarray hybridization, RNA was extracted from the tissue collected using the RNeasy Mini Kit (Qiagen), including a DNaseI step.

Complimentary RNAs (two rounds of amplification) were hybridized to Affymetrix GeneChip Mouse Genome arrays 430 v2, and the signal intensities were analysed using Partek Genomics suites (Partek, St Louis, MI, USA) and Matlab (The MathWorks Inc, Natick, MA, USA). The data were normalized using RMA and changes in gene expression $> 1.5$-fold with a $P$ value $< 0.05$ were considered to reflect a significant difference in expression.

**Purification of total RNA and quantitative real-time PCR.** VPM thalamic nucleus was dissected from coronal slices of 300 μm at E16.5. The total RNA was isolated using the kit NucleoSpin RNA (Macherey-Nagel, Düren, Germany), washed with the recommended buffers and eluted with RNase-free water by centrifugation. Quantification was done by optical density using a Nanodrop 2000 Spectrophotometer (Thermo Scientific, Wilmington, DE, USA) and the purity was evaluated by measuring the ratio of optical density at 260 and 280 nm. Complementary DNA (cDNA) was obtained from 1 μg of total RNA using the specific protocol for First-Strand cDNA synthesis in two-step reverse transcription–PCR (Thermo Scientific Random hexamer #SO142, RevertAid Reverse Transcriptase #EP0441, RiboLock RNase Inhibitor #EO0381, dNTP Mix #R0191) and stored at −20 °C. Quantitative PCR was performed in a StepOnePlus Real-Time PCR System (Applied Biosystems, Foster City, CA, USA) using the MicroAmp fast 96-well reaction plate (Applied Biosystems) and the Power SYBR Green PCR Master Mix (Applied Biosystems). The target *Rorβ* gene expression (Gene ID: 225998) was determined by the primers 5′-TTGTGGCGATAAATCCTCCG-3′ and 5′-TGCTGGCTCCTCCTGAAGAAT-3′. As a control we have used a housekeeping gene *Gapdh*, (Gene ID: 14433) determined by the primers 5′-CGG TGCTGAGTATGTCGTGGAGT-3′ and 5′-CGTGGTTCACACCCATCACAAA-3′. A master mix was prepared for each primer set containing the appropriate volume of SYBR Green, primers and template cDNA. All reactions were performed in duplicate as follows: 95 °C for 30 s and 45 cycles at 95 °C for 5 s and 60 °C for 35 s. The amplification efficiency for each primer pair and the cycle threshold (Ct) were determined automatically by the StepOne Software, v2.2.2 (Applied Biosystems). *Rorβ* transcript level was represented relative to the *Gapdh* signal adjusting for the variability in cDNA library preparation.

**Primary thalamic cell culture and transfection.** To establish primary thalamic neuron cultures, pregnant mice were killed and their embryos were recovered at E14.5. The thalamus was dissected, collected in Krebs solution, trypsinized and then dissociated with a fire-polished Pasteur pipette. In all, 200,000–300,000 cells per well were finally plated with plating medium (sodium pyruvate 1 mM, Glutamax 1 ×, 45% glucose solution 50 mM, penicillin/streptomycin 100 U per microlitre, 10% FBS, and 86% MEM). Thalamic neurons were transfected using Amaxa Basic Nucleofector Kit (VPI-1003) with 1 μg of the full-length *Rorβ* construct together with 1 μg of a plasmid encoding *Gfp*. Controls were transfected with 1 μg of a backbone construct together with 1 μg of *Gfp*. After transfection, cultures were placed for 10 days in maintenance medium (Glutamax 1 ×, 45% glucose solution 50 mM, penicillin/streptomycin 100 U per microlitre, 2% B27 and 95% neurobasal) at 37 °C with 5% $CO_2$. Maintenance medium was carefully replaced every 2 days.

**Quantification of axonal branches and axonal length.** ImageJ software (NeuronJ plugin) was used to analyse axon branching *in vitro*. Random fields were selected from three independent experiments. Axonal branches from neurons transfected with *i-Gfp* or *Rorβ-i-Gfp* were measured. Branches order criteria was determined by following the number of neurite bifurcations from the cell body. Total neurite length was determined by summing the lengths of all neurites for each neuron. For single axon reconstructions of VPM electroporated neurons *in vivo*, coronal sections of 60 μm were cut in a vibratome and single terminals of somatosensory TCAs were reconstructed in an individual barrel from the PMBSF using Neurolucida explorer from MBF Bioscience. The area occupied by the TCA terminals was measured by outlining the tips of each afferent with the ImageJ software.

**Calcium imaging in thalamocortical slices.** Pregnant adult mice were killed by decapitation after administering isofluorane, and their embryos were rapidly extracted and decapitated. Postnatal (P0-P3) and embryonic (E13.5-E18.5) mouse brains were immediately dissected out and kept in an ice-cold gassed slicing solution (95% $O_2$ and 5% $CO_2$) containing (in mM): 2.5 KCl, 7 Mg $SO_4$, 0.5 $CaCl_2$, 1 $NaH_2PO_4$, 26 $Na_2HCO_3$, 11 glucose and 228 sucrose. Oblique vibratome slices (300–350 μm thick: VT1200 Leica Microsystems Germany) were obtained along two axes: (1) $45 \pm 2°$ to preserve the somatosensory (VB) and visual (dLGN) thalamocortical nuclei and their connectivity; and (2) $-45 \pm 2°$ to preserve the auditory (MGv) and somatosensory (VB) thalamic nuclei. During the recovery period, the slices were placed at room temperature in standard aCSF (119 NaCl, 5 KCl, 1.3 Mg $SO_4$, 2.4 $CaCl_2$, 1 $NaH2PO4$, 26 $Na_2HCO_3$, 11 glucose) saturated by gassing with 95% $O_2$ and 5% $CO_2$.

For dye loading, the slices were incubated for 30–45 min in 2 ml of gassed aCSF (35–37 °C) with 10 μl Cal520 AM (AAT Bioquest) calcium dye (1 mM in DMSO + 20% pluronic acid). The loaded slices were left for 1 h at room temperature in gassed aCSF. The slices from $Gbx2^{CreER/+};R26^{GCaMP6-EGFP/+}$ mice

did not require incubation. Then, the slices were placed in a recording chamber of an upright Leica DM LFSA stage perfused (3.7 ml per minute) with warmed (32 °C) and gassed aCSF. Time lapse recording of $Ca^{2+}$ dynamics was obtained through water immersion objectives (L 10 × /0.30 N.A. Leica, L 20x/0.50 Leica) or dry objectives (HCX PL FLUOTAR 5 × /0.15 Leica, PL FLUOTAR 2.5 × /0.07) after exciting the slices at 492 nm with a mercury arc lamp. We acquired images with a digital CCD camera (Hamamatsu ORCA R2 C10600 10B), using an interframe interval of 250 ms and an exposure time of 200 ms. The length of each time lapse recording was 3,000 frames.

Pharmacological experiments were performed adding drugs in the warmed (32 °C), gassed (95% $O_2$ and 5% $CO_2$) and perfused (3.7 ml per minute) aCSF in the recording chamber. Carbenoxolone (Carbenoxolone disodium salt, Sigma) was applied at a concentration of 50 μM. Tetrodoxin (Abcam) was prepared at a concentration of 1 μM and mefloquine (Mefloquine Hydrochloride, Sigma-Aldrich) at 25 μM. High potassium concentration was applied by adding KCl to the aCSF (containing 5 mM of KCl) to reach 9, 10 and 12 mM concentrations. Recordings were performed 5 min after the start of drug application.

**Data analysis of calcium imaging.** Data were exported from Leica MM Fluor 1.6.0 acquisition software as 3,000 frame-long time-lapse sequences of TIFF format images and analysed in ImageJ and Matlab. Fluorescence traces of individual cells (recorded with L 20 × /0.50 objective) were analysed with custom developed routines in Matlab[67] and the calcium events were identified using asymmetric least square baselines and Schmitt trigger thresholds. We used 5% of the baseline noise as the upper threshold and 2% as the lower threshold. Onset and offset identification allowed raster plots to be generated and the extent of network correlation to be estimated. We define 'co-active' as meaning those onset transients occurring simultaneously within ± 1 frame time window (500 ms). The *correlation test* identified incidences where a significant fraction of the network's cells were active synchronously. The minimal size of these groups for each recording session was defined by the corresponding simulated random sets. For each of the 1,000 random data sets, we counted the number of co-active cells within each time-window and this defined a distribution of the expected counts of co-active cells due to random activity. The threshold for significant group co-activation was set as the 95th percentile of this distribution. Temporal colour-coding was performed using freely available Temporal-Colour Coder ImageJ plugin.

**Statistics.** Statistical analysis was carried out in GraphPad Prism6 and Matlab. Data are presented as mean and s.e.m. Statistical comparison between groups was performed using unpaired two-tailed Student's *t*-test or Mann-Whitney *U*-Test non-parametric two-tailed test when data failed a Kolmogorov-Smirnov or a Shapiro Wilk normality tests. For the whisker trimming experiments, a Two-way ANOVA test with Tukey *post hoc* analysis was used when interaction was not significant. For the drug application experiments Wilcoxon matched-pairs signed rank test was used. Simple effect analysis was performed when interaction was significant. $P$ values $< 0.05$ were considered statistically significant and set as follows *$P < 0.05$; **$P < 0.01$ and ***$P < 0.001$. No statistical methods were used to predetermine the sample size, but our sample sizes are considered adequate for the experiments and consistent with the literature. The mice were not randomized. The investigators were blinded to sample identity except in the calcium activity experiments.

**Data availability.** Microarray data has been deposited in the NCBI GEO database as GEO GSE76767.

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

## Acknowledgements

We are grateful M. Docquier and members of the NCCR genomics platform for their help with the expression arrays, to Denis Jabaudon for advice and help with the RNA extraction from thalamic structures, and to James Li for sharing the *Gbx2^CreER* mice. We are thankful to Oscar Marin, Angela Nieto, Paola Arlotta and members of G. López-Bendito's laboratory for stimulating discussions and comments. V.M.-J. holds a 'Severo Ochoa' PhD fellowship and N.A.-B. a FPI fellowship, both from the MINECO. C.M. held a JAE-Predoc fellowship from the CSIC, and H.G. held postdoctoral fellowships from the Swedish Research council and Brain Foundation. Supported by the Swiss National Science Foundation (31003A_149573) and the Novartis Research Foundation to F.M.R., the JSPS KAKENHI (JP16H06459) to T.I. and by the Spanish MINECO BFU2012-34298 and BFU2015-64432-R, and two European Commission Grants ERC-2009-StG-20081210 and ERC-2014-CoG-647012 to G.L.-B. G.L.-B. is an EMBO YIP Investigator and a FENS-Kavli scholar.

## Author contributions

G.L.-B. conceived the idea. V.M.-J. performed the experiments related to the embryonic enucleation mouse model, analysis of analysis of $Brn3b^{Cre}$;R26$^{tdTomato}$, $Ror\beta$ knockout and and $Nestin^{Cre}$;$Ror\beta^{fl}$ mice, and performed the dissociated thalamic cultures. N.A.-B.., performed the experiments related to the spontaneous activity in the MGv silenced mouse model. A.F. performed the $Ca^{2+}$ imaging experiments. C.M. and V.M.-J. performed the microarray assay. H.G., S.D. and F.M.R. generated the $R26^{Kir2.1-mCherry}$ mouse line. B.A. and V.M.-J. performed the $in\ utero$ electroporations. L.R.-M. performed mice perfusions and the $in\ situ$ hybridization experiments. R.S. genotyped the mouse colonies, generated the $in\ situ$ probes and plasmids for electroporation and performed the quantitative PCRs. O.S. analysed the microarray data. T.I. provided the TCA-GFP Tg mouse line. R.Sc. provided the $Ror\beta$ conditional mouse. M.R. and S.N. provided the $Ror\beta$ full knockout brains. V.M.-J., A.F. and N.A.-B. conducted the data analysis and M.V., F.M.R. and G.L.-B. wrote the paper.

## Additional information

**Competing financial interests:** The authors declare no competing financial interests.

