## [Peer Review File · Nature Communications]

Reviewers' comments:

Reviewer #1 (Remarks to the Author):

This is a very interesting and important study providing novel and further insights to central questions on cortical development. Over decades the role of intrinsic factors (e.g. transcription factors) versus external factors (e.g. synaptic inputs from the thalamus) have been intensely discussed in their role for cortical regionalization and arealization; protomap vs protocortex hypothesis (see for example seminal papers by Rakic and O'Leary supporting these different models).

In the current paper the authors use a variety of different state-of-the-art methods to study the role of spontaneous activity in the thalamus during early embryonic stages. For the first time spontaneous calcium waves in specific thalamic nuclei of different sensory systems could be observed as early as E14.5. These waves show a very interesting specific pattern of propagation among three sensory-modality nuclei. Using a number of different experimental approaches the authors convincingly show that these thalamic waves regulate the cortical representations. As the underlying mechanisms controlling these spontaneous activity patterns the authors identified the nuclear orphan receptor Ror-beta in the thalamus as a key regulator of cortical area formation.

I find this study very convincing. The data are completely novel and are most important for everybody interested in cortical development. It was a pleasure to read this paper. I have only a few questions and comments, which may be helpful to improve some aspects of this interesting study.

1) At early stage the cortical plate is very immature and the thalamic input reaches the cortex via the subplate. A number of studies (although in older rodents at P0) have shown that the subplate may actively transmit the thalamic activity to the developing neocortical network (e.g. Nature 439: 79ff). When and how does this thalamic activity spread to the cortex? Is the activity in the sensory cortex restricted to one area (e.g. V1)?

2) What are the mechanisms of wave propagation? Glia? The authors exclude a role of astrocytes. Maybe from the velocity of spread the authors may argue for a mechanism. Wave of potassium, glutamate or GABA release?

3) What defines the surprisingly clear border of wave termination in the thalamus (within dLGN, VPM and MGv)?

4) Is there a trigger zone within one thalamic nucleus from where the activity often starts?

5) All gap junction blockers, including carbenoxolone, are more or less unspecific and we have no better tools, but the authors may want to try mefloquine, which has been successfully used in the past to block Cx36 containing gap junctions.

6) Did the authors try to record local field potentials in the thalamus in combination with the imaging? LFP recordings are easy to perform, provide a higher temporal resolution and may give some further information on the properties of the spontaneous activity.

7) Is Ror-beta expressed in the subplate at early stages?

Reviewer #2 (Remarks to the Author):

In this manuscript by Moreno-Juan et al. the authors show that: 1. gap junction derived waves of activity travel between sensory nuclei (MG, LGN, and VPM) in the developing thalamus (at least in slices). 2. Altering retinal inputs into the dLGN or silencing neurons in the MG alter the dynamic of these waves, leading to an increase of frequency of waves in the VPM 3. These changes also lead to an increase of RorB expression in the VPM. 4. Loss of retinal inputs, silencing of the MG, or an increase of RorB expression leads to an expansion of S1, and loss of RorB expression leads to a decrease in S1 area. 5. Neurons that over express RorB have increased branching and dendritic length. From these results the authors posit a model whereby the spatial temporal pattern of calcium waves in the thalamus regulate cortical area size prior to sensory processing via induction of RorB. These conclusions are original and of interest to neuroscientists that study how brain wiring develops. The approach is valid and the data is of good quality. I have two concerns and a few minor comments that if addressed would greatly improve the value and significance of this work.

1. While the slice data are compelling the significance of this work would be increased substantially if the authors could show that the thalamic waves happen in vivo.

2. In Figure 6 the authors present a model whereby enucleation or silencing MG leads to an increase in size of S1 at the expense of V1 or A1. It is important that the authors demonstrate this. Do V1 and A1 decrease in area under these conditions?

Comments

1. A microarray experiment identified 106 genes that change expression upon enucleation. What are these genes? There is no list given. It seems that many genes had a greater change in gene expression than RorB. Why were these genes not verified?

2. Jabaudon et al. 2012 showed that RorB expression in the cortex is regulated by TC axons. They also showed that ectopic expression of RorB in the cortex leads to alterations in barrel structure. Therefore it is important that the thalamic changes seen in RorB are specific for the VPM. In Figure 4D it appears that other areas of the thalamus also have increased expression in the enucleated mice. If the thalamic waves were driving the expression change seen in the VPM wouldn't you expect the upregulation to be VPM specific? It is also imperative that the authors also show a control gene that is not upregulated upon enucleation, it would also be nice to add a panel showing a section from the RorB mutant control.

In the same vein the overexpression of RorB in figure 5 needs to be restricted to the thalamus. This is not obvious from figure S6.

In the Discussion (line 320) the authors say "Our data suggest that prenatal thalamic Ror β expression is a main regulator of the size of the barrel cortex representation". This is misleading as RorB mutants only change the size by 10%, therefore something else must be the main regulator of size.

Reviewer #3 (Remarks to the Author):

In this interesting manuscript, Moreno-Juan, Filipchuk and collaborators reveal the existence of prenatal calcium waves which propagate across sensory thalamic nuclei and investigate the role of these waves in regulating the size of cortical areas.

The authors start off by showing that S1 cortical area is increased following prenatal enucleation; they follow by revealing the existence of prenatal calcium waves propagating across the VPM, dLGN and MGv. They next manipulate these waves in the MGv using Kir oe to show that this is associated with an increased size of S1, then switch back to the enucleation model to investigate gene expression in

the VPM, identifying RORB as a differentially regulated gene, and finally perform gain-and-loss-of functions to investigate the role of RORB in S1 size.

While the finding of intra-thalamic calcium waves is novel and constitutes a fundamental contribution to our understanding of the assembly and plasticity of sensory pathways, I am less convinced by other claims of the manuscript, for which experimental evidence is insufficient or lacking. In particular, the relationship between calcium waves, *rorb* expression, and S1 size is loosely presented, and current evidence does not support the claim that intracortical calcium waves act pre-cortically to regulate the balance between cortical areas. I would thus suggest that the authors focus on their key finding (intra-thalamic calcium waves), and in fact open the paper with figure 2, while using the rest of their findings as illustrations of the potential functional role of these waves. The finding that the waves are mediated through gap junctions is fundamental and would deserve to be upgraded to the main figures.

Despite these significant shortcomings, this is a groundbreaking study that I feel deserves publication in Nature Communication, provided the points below are addressed:

1. I am not convinced that the expansion of S1 following enucleation is specific to this sensory area, or in fact restricted to sensory areas. Could it not represent a global expansion of all areas (including non-sensory areas) secondary to the (assumed) shrinkage of V1? Along the same line of thought, it seems like the expansion of S1 is restricted to one axis of this area (mostly medio-lateral), while the antero-posterior expansion is not obviously affected. Regarding the first point, the authors should assess the overall distortion of cortical areas using area-specific markers, including for non-sensory areas (and also show shrinkage of V1). Regarding the second point, quantification across these two axes should be performed and discussed.
2. In Fig.1 D, the authors quantify the size of individual barrels, yet the number of barrels presented is small compared to the overall number of S1 quantified in 1C. All barrels of all S1 should be presented, and values for each barrel (C1,C2 etc) should be given as mean \pm SEM in this and similar figures.
3. Is the volume of the VPM modified following enucleation? What about the size of the dLGN and the MGv? Fig. S5a seems to suggest that the VPM is larger; this should be carefully quantified. The authors' general claim is that changes in S1 size occur in the absence of any morphological changes at the level of the VPM; this need to be precisely assessed in the enucleation, Kir, and RORB paradigms.
4. Assessment of rewiring is not adequately performed in Fig. S5a. diI and diA are essentially anterograde labeling dyes and the current experiments thus mostly illustrate top-down connectivity rather than thalamic output. These dyes should be placed in the distinct sensory nuclei or a genuine retrograde labeling approach should be used.
5. In figure 2a and other figures, it seems like the waves significantly extend into the P0m. This should be mentioned. Are other thalamic nuclei also activated?
6. To confidently claim that calcium waves in the VPM regulates the size of the barrel field in the cortex, the authors need to demonstrate that abolishing calcium waves in the VPM causes a decreased size of the PMBSF in S1. Can the authors not perform this with the Kir approach? The authors should distinguish instructive vs. permissive roles of calcium waves in their discussion (see e.g. papers from the Crair lab). Unless this experiment is performed (and leads to conclusive results), the causal relationship between calcium waves and area size remains circumstantial, and claims relating to this topic should be toned down.
7. The authors cannot exclude that the observed trans-modal plasticity effects are the result of cortico-cortical interactions rather than thalamo-thalamic. Similarly, thalamo-cortico-thalamic effects have not been investigated. The title and overall text should thus be toned down to acknowledge these alternative possibilities. Of note, despite its title, the findings in the Zembryski et al paper does not demonstrate a top-down action onto the somatosensory cortex but could in fact result from lack of stabilization of TC axons due to inadequate targets.
8. The list of differentially-expressed genes following enucleation should be provided as a

supplementary table to better document the transcriptional networks activated by this procedure and the position of RORB in this context.

9. How is expression of RORB in the MGv and dLGN modified by Kir and enucleation, respectively? Is RORB not affected in the MGv following enucleation?

10. To support the claim that RORB is involved in the change in S1 area size occurring following enucleation or Kir oe, it is absolutely essential that the authors examine S1 following enucleation or Kir oe in RORB KO mice. The arborization of TC axons should be carefully assessed (see 12).

11. The limitations of the RORB KO mice should be discussed in detail (i.e. full KO as opposed to thalamic-specific). RORB is also expressed in L4 cortical neurons; how does this affect interpretation of the results?

12. What does area or barrel size actually mean? What is the anatomical substrate of this increase in size? The authors should carefully assess the axonal arborization of individual VPM TC axons in vivo to address this important question. Fig. 5e is not convincing at all; how many mice have been electroporated? Some barrels seem to contain very little RORBi GFP and yet have different sizes compared to controls. This should also be illustrated in coronal sections with careful quantifications.

13. How do the authors reconcile the prenatal occurrence of the waves with postnatal changes in gene expression? Previous work has shown that S1 size is affected by enucleation postnatally, how do the current findings relate to this? What effect does a late postnatal enucleation have on S1 size?

Exclusion of postnatal input-dependent processes is critical in this sense, and postnatal trimming is not sufficient to exclude this possibility (skin receptors remain activatable by direct contact). Cautery of whisker follicles would be necessary to exclude this possibility. This should, at least, be discussed.

14. I could not find the Legends for Supplementary Figures.

Point-by-point Response to Reviewers' Comments (NCOMMS-16-16239-T)

Reviewer #1

"This is a very interesting and important study providing novel and further insights to central questions on cortical development. Over decades the role of intrinsic factors (e.g. transcription factors) versus external factors (e.g. synaptic inputs from the thalamus) have been intensely discussed in their role for cortical regionalization and arealization; protomap vs protocortex hypothesis (see for example seminal papers by Rakic and O'Leary supporting these different models)...I find this study very convincing. The data are completely novel and are most important for everybody interested in cortical development. It was a pleasure to read this paper. I have only a few questions and comments, which may be helpful to improve some aspects of this interesting study."

Response: We thank the reviewer for considering our data "very convincing and very interesting", and for providing valuable and constructive comments.

Point 1. "At early stage the cortical plate is very immature and the thalamic input reaches the cortex via the subplate. A number of studies (although in older rodents at P0) have shown that the subplate may actively transmit the thalamic activity to the developing neocortical network (e.g. Nature 439: 79ff). When and how does this thalamic activity spread to the cortex? Is the activity in the sensory cortex restricted to one area (e.g. V1)?"

Response: We thank the reviewer for this comment. We also consider the transmission of thalamic waves to the cortex an interesting possibility. To address this issue, we have generated a transgenic mouse in which the calcium indicator GCaMP6 is expressed in thalamic neurons ($Gbx2^{CreER} \times R26^{GCaMP6}$) and in their projections to the cortex. By performing calcium imaging in the thalamocortical fibres at the level of the ventral telencephalon and the neocortex we found that thalamic calcium waves are transmitted to the cortex reaching distinct medio-lateral areas (new data included in **Figure 2e-g**).

Point 2. "What are the mechanisms of wave propagation? Glia? The authors exclude a role of astrocytes. Maybe from the velocity of spread the authors may argue for a mechanism. Wave of potassium, glutamate or GABA release?"

Response: We agree with the reviewer that the mechanisms of wave propagation deserve further investigation. We have started to characterize the underlying mechanisms by performing pharmacological experiments in combination with calcium imaging in acute slices. The addition of the voltage-dependent sodium channel blocker, TTX, completely abolished the thalamic waves without substantially affecting the inter-waves activity (**Figures 3d and 3e**). Furthermore, we performed experiments in which the membrane voltage is manipulated by increasing the extracellular potassium concentration. Our data shows that membrane depolarization increases the frequency of the thalamic waves in a dose-dependent manner (**Figure 3f**). Altogether these results suggest that the mechanism of thalamic waves initiation/propagation depend on changes on membrane voltage and the participation of gap junctions. Still, a full characterization of the mechanisms involved in the initiation and spread of the waves will require deeper investigation. However, we think that such characterization is beyond the scope of the present study. Finally, we have performed an immunocytochemistry against the glial markers GFAP and S100 β ($n = 3$ at E16.5, $n = 2$ at P30) and found no labeling at the thalamus at E16.5 (**Figure 1 for Reviewers**), suggesting that it is unlikely that glial cells participate in this feature. These new results have been commented in the Discussion (**page 16**).

Figure 1 for Reviewers

Point 3. “What defines the surprisingly clear border of wave termination in the thalamus (within dLGN, VPM and MGv)?”

Response: This is an interesting and very robust property of the waves. Our pharmacological studies with the gap junction blocker carbenoxolone strongly suggest that gap-junctions are part of the mechanism involved in the propagation of the thalamic waves. We hypothesize that connexins, which are the anatomical substrate for the gap junctions, might be differentially expressed in the distinct thalamic territories. To check this possibility, we have performed *in situ* hybridization for connexin-36 and connexin-43. Connexin-36 showed a weak and uniform pattern of expression across thalamic territories without delineating clear anatomical boundaries, which is consistent with our new experiments showing that in the presence of Mefloquine (a Cx36 blocker), thalamic waves persist (**Figure S4i**). Connexin-43 is expressed in astrocytes and radial glia during development (Cotrina et al., 1998) and we found very weak expression in the thalamus prenatally. In this sense, our preliminary data obtained from a connexin-43 knockout mouse show that waves persist with the same frequency, in the absence of this connexin (data not shown). Thus, our working hypothesis is that the limits of the waves depend on the activity of several connexins. Perhaps, Cx45 is a good candidate as it shows a specific pattern of expression in the principal thalamic nuclei later on at the postnatal life (Söhl et al., 2005). In this regard, it has been detected the presence of connexins-dependent electrical synaptic coupling between pairs of VPM cells at postnatal stages, which decreases over time (Lee et al 2010). However, there is not information about this parameter during the embryonic development of the different thalamic nuclei. Thereby, in the absence of specific pharmacological tools, it would be necessary to measure the electrical coupling coefficient between pairs of embryonic thalamic neurons. A difference in this coefficient might explain the borders defining the termination of the waves.

Point 4. “Is there a trigger zone within one thalamic nucleus from where the activity often starts?”

Response: We thank the reviewer for raising this point. As shown in **Figure 3a**, we found that independently of their origin, waves constantly cover the same territory respecting the boundaries of regions that do not show waves. Moreover, we have found that there are nuclei with more capacity of generation of waves, notably the VPM (**Figure 3b**). To clarify this point, we have plotted the

distribution of waves origins quantified per pair of nuclei (dLGN-VPM, $n = 40$; and MGv-VPM, $n = 61$). As shown in **Figure 3c**, there is a uniform distribution of origins and we could not detect a “trigger” or “hot” area within a nucleus.

Point 5: “All gap junction blockers, including carbenoxolone, are more or less unspecific and we have no better tools, but the authors may want to try mefloquine, which has been successfully used in the past to block Cx36 containing gap junctions”

Response: We thank the reviewer for this suggestion. We agree that there are no good tools to specifically block gap junctions. However, carbenoxolone has been widely used as a general gap junction blocker (Niculescu & Lohmann, 2013). Nevertheless, as suggested by the reviewer we have performed additional experiments using mefloquine ($n = 3$). Our results show that Mefloquine does not completely block the thalamic calcium waves (**Figure S4i**), as carbenoxolone does, suggesting that Cx36 might be only partially involved in the mechanism of waves propagation.

Point 6: “Did the authors try to record local field potentials in the thalamus in combination with the imaging? LFP recordings are easy to perform, provide a higher temporal resolution and may give some further information on the properties of the spontaneous activity”

Response: This is an interesting suggestion by the reviewer. We have been recording LFP not in combination with calcium-imaging but as the initial approach to record spontaneous thalamic activity in newborn mice *in vivo*. The rationale for the experiments and the preliminary results are shown in the response to point 1 of Reviewer#2 (**Figure 3 for Reviewers**).

Point 7: “Is Ror-beta expressed in the subplate at early stages?”

Response: We have now checked this possibility and found that *Rorβ* is not expressed in the subplate at either E16.5 ($n = 3$) or P0 ($n = 3$) (**Figure 2 for Reviewers**).

Figure 2 for Reviewers

Reviewer #2

“These conclusions are original and of interest to neuroscientists that study how brain wiring develops. The approach is valid and the data is of good quality. I have two concerns and a few minor comments that if addressed would greatly improve the value and significance of this work”

Response: We thank the reviewer for considering our data “original and of good quality”, and for providing valuable and constructive comments.

Point 1: “While the slice data are compelling the significance of this work would be increased substantially if the authors could show that the thalamic waves happen in vivo”

Response: We completely agree with the reviewer that recording the existence of thalamic waves *in vivo* is certainly the next step. During the last year we have been pursuing this aim, however this has proven to be a very challenging experiment. There are two main reasons. First, the thalamus is a deep brain structure of difficult accessibility for calcium imaging experiments. Second calcium waves need to be recorded at very early stages, from E16.5 to P0, meaning that the experiments need to be done in live fetuses. We have set up a method for *in vivo* recordings of mouse fetuses that are connected to the anaesthetized mother via the intact umbilical cord. Recently, we have developed a transgenic mouse in which the calcium indicator GCaMP6 is expressed in thalamocortical neurons ($Gbx2^{CreER} \times R26^{GCaMP6}$). As indicated in the response to a previous point (Reviewer 1; point#1), in these animals we have been able to record the spread of the thalamic waves to the neocortex in slices (new results described in **Figure 2e-g**); therefore, we are confident about the feasibility of these experiments in the *in vivo* fetuses. However, our data is too preliminary at the moment. **[UNPUBLISHED DATA REDACTED BY EDITORIAL TEAM]**.

Point 2: “In Figure 6 the authors present a model whereby enucleation or silencing MG leads to an increase in size of S1 at the expense of V1 or A1. It is important that the authors demonstrate this. Do V1 and A1 decrease in area under these conditions?”

Response: We thank the reviewer for raising this important point. We have now quantified the changes in size in the remaining principal sensory cortical areas in both embBE and MGV^{Kir} models. To this end and to better visualize the principal cortical areas, we have performed the embryonic enucleations or the MGv silencing in a background mouse in which TCAs are labeled by GFP (TCA-GFP mouse). After embBE, we have found a reduction of 33.3% of the V1 area that is accompanied by a similar reduction of the dLGN size (39.8%) at P8 (**Figure 1b-c** and **Figure S1**). The A1 and MGv areas are not changed after embBE (**Figure 1b-c** and **Figure S1**). It is more than likely that the reduction in both dLGN and V1 size in the embBE are due to a lack of a trophic effect by the absence of retinal axons in this model. This effect has been previously reported in mice and other animal models after early postnatal enucleation (Dehay et al., 1989; Karlen & Krubitzer, 2009; Kozanian et al., 2015).

In the MGV^{Kir} model, we found an enlargement of the S1 territory but also a concomitant increase of 14.6% of the V1 area and a 51.9% reduction of the A1 territory at P7 (**Figure S6**). None of the principal thalamic nuclei (VPM, dLGN and MGv) are changed in size in the MGV^{Kir} mice (**Figure S6**). In this mouse, we also found that $Ror\beta$ expression is decreased in the MGv nucleus, opposite to what we found in the VPM of the MGV^{Kir} and embBE mice. Thus, it is tempting to speculate that the lack of waves in the MGv nucleus leads to a reduction of A1 in the MGV^{Kir} mouse probably through decreasing TCA axonal terminal branching. The mechanism by which V1 area is expanded in the MGV^{Kir} mouse seems to be independent on $Ror\beta$.

Comments

Point 1. “A microarray experiment identified 106 genes that change expression upon enucleation. What are these genes? There is no list given. It seems that many genes had a greater change in

gene expression than *RorB*. Why were these genes not verified?"

Response: All the microarray data we have generated here is provided openly to the community with the publication of this manuscript by the deposit in the GEO database as indicated in **page 31**. Nevertheless, we agree with the reviewer that providing a list of the most significantly changed genes could be helpful in the main text. We have now listed the top ten significantly changed genes in **Figure 5b**. Among these, *Rorβ* is number six from the top of the most significantly upregulated genes in the VPM. We concentrated in the analysis in *Rorβ* because of its previous implication in barrel cortex formation (Jabaudon et al., 2012).

Point 2. "Jabaudon et al. 2012 showed that RorB expression in the cortex is regulated by TC axons. They also showed that ectopic expression of RorB in the cortex leads to alterations in barrel structure. Therefore it is important that the thalamic changes seen in RorB are specific for the VPM. In Figure 4D it appears that other areas of the thalamus also have increased expression in the enucleated mice. If the thalamic waves were driving the expression change seen in the VPM wouldn't you expect the upregulation to be VPM specific? It is also imperative that the authors also show a control gene that is not upregulated upon enucleation, it would also be nice to add a panel showing a section from the RorB mutant control"

Response: We thank the reviewer for raising this point. We mentioned in the results section that these thalamic waves exist from E14.5 until P2. Although from E14.5 to E17 thalamic waves are only present in the principal thalamic nuclei (dLGN, VPM and MGv), later on, secondary nuclei (as the LP or the P_{Om}) are also engaged in the waves phenomenon. We have now included this information in the result section (**page 7**). Thus, we do not exclude the possibility that at P0, when we describe by ISH the upregulation of *Rorβ* in the VPM, other areas of the thalamus might also show a similar upregulation. Following the reviewer's suggestion, we have now included a control gene, cholecystokinin (Cck), which is enriched in the embryonic VPM (Gezelius et al., 2016) and that did not show a change in expression in the Microarray data. The expression of this gene did not change in the VPM of the embBE mice as shown in **Figure S8**.

"In the same vein the overexpression of RorB in figure 5 needs to be restricted to the thalamus. This is not obvious from figure S6"

Response: We apologize if this point was not clear enough. We verified that all the brains included for quantification of the S1 size at P6 were selectively electroporated into the thalamus and that the VPM was targeted. For this purpose, cortices were cut tangentially and the diencephalon from the same brain and hemisphere was cut coronally. Electroporations that targeted non-desired regions outside the thalamus or did not target the VPM were excluded from the quantification. Moreover, we corroborated the efficiency and specificity of the electroporation by checking the targeting to the thalamus at E18.5 (now included in **Figure S10**).

"In the Discussion (line 320) the authors say "Our data suggest that prenatal thalamic Rorβ expression is a main regulator of the size of the barrel cortex representation". This is misleading as RorB mutants only change the size by 10%, therefore something else must be the main regulator of size"

Response: We agree with the reviewer that other factors may also contribute to the mechanisms of regulation of cortical areas size. We have toned down this statement accordingly. On the other hand, in further support of the important role of *Rorβ*, we have now found that the size of S1 is reduced by 18.1% when using a *Nestin^{Cre};Rorβ^f* conditional approach (new data included in **Figure 6**).

Reviewer #3

*“While the finding of intra-thalamic calcium waves is novel and constitutes a fundamental contribution to our understanding of the assembly and plasticity of sensory pathways, I am less convinced by other claims of the manuscript, for which experimental evidence is insufficient or lacking. In particular, the relationship between calcium waves, *Rorb* expression, and S1 size is loosely presented, and current evidence does not support the claim that intracortical calcium waves act pre-cortically to regulate the balance between cortical areas. I would thus suggest that the authors focus on their key finding (intra-thalamic calcium waves), and in fact open the paper with figure 2, while using the rest of their findings as illustrations of the potential functional role of these waves. The finding that the waves are mediated through gap junctions is fundamental and would deserve to be upgraded to the main figures. Despite these significant shortcomings, this is a groundbreaking study that I feel deserves publication in Nature Communication, provided the points below are addressed”*

Response: We thank the reviewer for considering our finding “novel and that constitutes a fundamental contribution” and for providing valuable and constructive comments. We believe we have provided novel data from additional experiments that strengthen the links between the calcium waves, thalamic *Rorb* expression and the cortical areas size modifications as described below.

Point 1. “I am not convinced that the expansion of S1 following enucleation is specific to this sensory area, or in fact restricted to sensory areas. Could it not represent a global expansion of all areas (including non-sensory areas) secondary to the (assumed) shrinkage of V1? Along the same line of thought, it seems like the expansion of S1 is restricted to one axis of this area (mostly medio-lateral), while the antero-posterior expansion is not obviously affected. Regarding the first point, the authors should assess the overall distortion of cortical areas using area-specific markers, including for non-sensory areas (and also show shrinkage of V1). Regarding the second point, quantification across these two axes should be performed and discussed”

Response: We completely agree with the reviewer that this is an important issue. We have now quantified the changes in size in the remaining principal sensory cortical areas in both embBE and *MGv^{Kir}* models. To this end and to better visualize the principal cortical areas, we have performed the embryonic enucleations or the *MGv* silencing using as a genetic background a mouse line in which TCAs are labeled by GFP (TCA-GFP, Mizuno et al., 2014). After embBE, we have found a reduction of 33.3% of the V1 area that is accompanied by a similar reduction of the dLGN size (39.8%) at P8 (**Figure 1b-c** and **Figure S1**). The A1 and *MGv* areas are not changed after embBE (**Figure 1b-c** and **Figure S1**). It is more than likely that the reduction in both dLGN and V1 size in the embBE are due to a lack of a trophic effect by the absence of retinal axons in this model. This effect has been previously reported in mice and other animal models after early postnatal enucleation (Dehay et al., 1989; Karlen & Krubitzer, 2009; Kozanian et al., 2015).

In the *MGv^{Kir}* model, we found an enlargement of the S1 territory but also a 14.6% of the V1 area and a 51.9% reduction of the A1 area at P8 (**Figure S6**). None of the principal thalamic nuclei (VPM, dLGN and *MGv*) are changed in size in the *MGv^{Kir}* mice (**Figure S6**). In this mouse, we also found that *Rorb* expression is decreased in the *MGv* nucleus, opposite to what we found in the VPM of the *MGv^{Kir}* and embBE mice. Thus, it is tempting to speculate that the lack of waves in the *MGv* nucleus leads to a reduction of A1 in the *MGv^{Kir}* mouse probably through decreasing TCA axonal terminal branching. The mechanism by which V1 area is expanded in the *MGv^{Kir}* mouse seems to be independent on *Rorb*.

Regarding the second point mentioned by the reviewer, we have quantified in controls and embBE mice the expansion of the PMBSF along the medio-lateral axis (630.30 ± 39.04 pixels in control and 670.71 ± 38.79 pixels in embBE) and the antero-posterior axis (326.60 ± 18.66 in control and 337.93 ± 26.35 in enucleated animals). The ratio of the medio-lateral/antero-posterior axis in control and enucleated animals is not different (1.94 ± 0.17 and 1.99 ± 0.15 , respectively) suggesting that the growth is proportional in both axes. These data have been included in **Figure 1**

legend.

Point 2. "In Fig.1 D, the authors quantify the size of individual barrels, yet the number of barrels presented is small compared to the overall number of S1 quantified in 1C. All barrels of all S1 should be presented, and values for each barrel (C1,C2 etc) should be given as mean \pm SEM in this and similar figures"

Response: We apologize for the lack of clarity regarding this point. A total of $n = 14$ barrels were quantified at P4 and $n = 19$ barrels at P7, P8 and P30. The identity of the barrels quantified at each stage is the same for each animal analyzed. For the quantification of the individual barrels in the *Rorb* KO, a total of $n = 12$ barrels were quantified. Following the reviewer comment, we have now updated the quantification of the size of individual barrels to show the mean \pm SEM for each barrel and condition in each Figure. Furthermore, we have listed the identity of the barrels that are significantly changed for each animal model and condition (**Figure 1**, **Figure 6** and **Figure S2**).

Point 3. "Is the volume of the VPM modified following enucleation? What about the size of the dLGN and the MGv? Fig. S5a seems to suggest that the VPM is larger; this should be carefully quantified. The authors' general claim is that changes in S1 size occur in the absence of any morphological changes at the level of the VPM; this need to be precisely assessed in the enucleation, Kir, and RORB paradigms"

Response: We agree with the reviewer that these quantifications should be provided. We have now quantified the changes in area and volume of the principal thalamic nuclei (dLGN, VPM, MGv) in the embBE and MGv^{Kir} models (**Figure S1** and **Figure S6**). Only the dLGN is significantly decreased in the embBE mice probably due to a lack of a trophic effect from the absence of retinal axons in this model. The reduction in the dLGN size after early enucleations has been previously demonstrated (Dehay et al., 1989; Karlen & Krubitzer, 2009; Kozanian et al., 2015). In the MGv^{Kir}, none of the principal sensory thalamic nuclei change in size.

Point 4. "Assessment of rewiring is not adequately performed in Fig. S5a. dil and diA are essentially anterograde labeling dyes and the current experiments thus mostly illustrate top-down connectivity rather than thalamic output. These dyes should be placed in the distinct sensory nuclei or a genuine retrograde labeling approach should be used"

Response: We have now performed anterograde labeling by Dil injections from the distinct sensory thalamic nuclei. The data shows no rewiring of TCAs in the targeting of the primary cortical areas. This data has now been included in **Figure S7**.

Point 5. "In figure 2a and other figures, it seems like the waves significantly extend into the POm. This should be mentioned. Are other thalamic nuclei also activated?"

Response: We thank the reviewer for raising this point. We mentioned in the results section that these thalamic waves exist from E14.5 until P2. Although from E14.5 to E17 thalamic waves are only present in the principal thalamic nuclei (dLGN, VPM and MGv), later on, secondary nuclei (as the LP or the POm) are also engaged in the waves phenomenon. We have now included this information in the Results section (**page 7**).

Point 6. "To confidently claim that calcium waves in the VPM regulates the size of the barrel field in the cortex, the authors need to demonstrate that abolishing calcium waves in the VPM causes a decreased size of the PMBSF in S1. Can the authors not perform this with the Kir approach? The authors should distinguish instructive vs. permissive roles of calcium waves in their discussion (see e.g. papers from the Crair lab). Unless this experiment is performed (and leads to conclusive

results), the causal relationship between calcium waves and area size remains circumstantial, and claims relating to this topic should be toned down”

Response: We understand the reviewer comment. Unfortunately, the conditional targeting of the distinct thalamic nuclei by the *Gbx2* expression does not allow (at present) to exclusively target the VPM without targeting the dLGN and the MGv nuclei as well. This is due to the fact that there is not a window of time where VPM neurons exclusively express *Gbx2*. MGv neurons always express *Gbx2* and the expression of *Gbx2* in the dLGN neurons virtually overlaps with that of VPM neurons. We agree with the reviewer that an instructive vs. permissive role of calcium waves should be discussed and we have now included a paragraph on this issue in the Discussion section (**page 16**). Nevertheless, the fact that in the *MGV^{Kir}* mice the abolishment of the thalamic calcium waves in the MGv correlates with a decrease in the size of the A1 area (new data in **Figure S6**) strongly suggest an instructive role of these waves in the developmental program of cortical areas.

Point 7. “The authors cannot exclude that the observed trans-modal plasticity effects are the result of cortico-cortical interactions rather than thalamo-thalamic. Similarly, thalamo-cortico-thalamic effects have not been investigated. The title and overall text should thus be toned down to acknowledge these alternative possibilities. Of note, despite its title, the findings in the Zembrzyski et al paper does not demonstrate a top-down action onto the somatosensory cortex but could in fact result from lack of stabilization of TC axons due to inadequate targets”

Response: We completely agree with the reviewer that we cannot exclude the possibility of a cortico-cortical influence in addition to the thalamocortical mechanism described here. The fact that silencing the waves in the MGv leads to an increase in S1 size and a decrease in A1 size, without affecting the thalamic nuclei territories, suggest that the top-down plasticity described in Zembrzyski et al might play a minor role in this scenario. Nevertheless, we have acknowledged other possibilities in the Discussion section (**page 15**) and generally toned down the text.

Point 8. “The list of differentially-expressed genes following enucleation should be provided as a supplementary table to better document the transcriptional networks activated by this procedure and the position of RORB in this context”

Response: All the microarray data we have generated here is provided openly to the community with the publication of this manuscript by the deposit in the GEO database as indicated in **page 31**. Nevertheless, we agree with the reviewer that providing a list of the most significantly changed genes could be helpful in the main text. We have now listed the top ten significantly changed genes in **Figure 5b**. Among these, *Rorb* is number six from the top of the most significantly upregulated genes. We concentrated in *Rorb* because of its previous implication in barrel cortex formation (Jabaudon et al., 2012).

Point 9. How is expression of RORB in the MGv and dLGN modified by Kir and enucleation, respectively? Is RORB not affected in the MGv following enucleation?

Response: We thank the reviewer for raising this point. *Rorb* is very weakly expressed in the dLGN at these early developmental stages. Nevertheless, we performed ISH for *Rorb* in both embBE and *MGV^{Kir}* mice and found that whereas *Rorb* is not changed in the MGv of the embBE mice at P0 (**Figure S8**), it is downregulated in the MGv of the *MGV^{Kir}* mouse (**Figure S8**), where A1 area is also reduced.

Regarding its regulation, we have strengthened the link on the activity-dependent regulation of *Rorb* by performing a qPCR in control VPM neurons or in VPM neurons after treatment with a high concentration of KCl. Elevated levels of KCl increase the frequency of the thalamic calcium waves (**Figure 3f**), which is correlated with an upregulation of *Rorb* levels in the VPM (**Figure 5e**).

Point 10. “To support the claim that RORB is involved in the change in S1 area size occurring following enucleation or Kir oe, it is absolutely essential that the authors examine S1 following enucleation or Kir oe in RORB KO mice. The arborization of TC axons should be carefully assessed (see 12)”. . . . “The limitations of the RORB KO mice should be discussed in detail (i.e. full KO as opposed to thalamic-specific). RORB is also expressed in L4 cortical neurons; how does this affect interpretation of the results?”

Response: Our findings with gain-of-function experiments of *Rorb* in the thalamus *in vivo* strongly support a role of *Rorb* in the change in S1 area. Nevertheless, to strengthen this point, we have performed additional experiments. First, we have assessed the arborization of single thalamocortical axons overexpressing *Rorb* in the barrel field of S1 and showed that there is a significant increase in the arborization of thalamic axon terminals under these conditions compared to controls. These data further support our previous *in vitro* and *in vivo* results and have now been included in **Figure 6**. Although our loss-of-function experiments in a *Rorb* KO showed shrinkage of S1 in the absence of this gene, we agree that the n was low. We have nicely and strongly reproduced the shrinkage of S1 by using a *Nestin^{Cre};Rorb^{fl/fl}* conditional approach that we set in our laboratory (data now included in **Figure 6**). In this model, *Rorb* is deleted in all neurons.

We completely understand that performing an embBE or Kir-silencing in a *Rorb*-deficient background could, if successful, strengthen our results. We have heavily tried to get this experiment done by performing embBE in the conditional *Nestin^{Cre/+};Rorb^{fl/fl}* mice. However, several problems emerged that complicated the viability of the experiment. The *Nestin^{Cre};Rorb^{fl/fl}* mice are in a C57 background that is weaker and has a reduced number of embryos compared to the ICR background in which all the embBE experiments were performed. Very few mutants, bilaterally enucleated, survive up to P8. We could only get $n = 2$ *Nestin^{Cre/+};Rorb^{fl/fl}* survivors at P8 in the last 3 months. However, the effect was clear and those mutants showed the reduction of S1 size (**Figure 4 for Reviewers**). This preliminary result suggests that, in the absence of *Rorb*, the mechanism of cross-modal enlargement of S1 triggered by the embBE does not occur and further support a central role for *Rorb* in this process. As this data is too preliminary, we have not included it into the manuscript.

Finally, as mentioned by the reviewer, we completely agree that using a thalamic-specific conditional deletion of *Rorb* would be ideal. However, as aforementioned in the response to Point#6, the selective embryonic conditional targeting of VPM neurons is at present not possible using the *Gbx2-CreER* mouse. To our knowledge, there is no other tool available that we could use to embryonically and exclusively target the VPM to assess the function of *Rorb* in this context.

Figure 4 for Reviewers

Point 11. "What does area or barrel size actually mean? What is the anatomical substrate of this increase in size? The authors should carefully assess the axonal arborization of individual VPM TC axons in vivo to address this important question. Fig. 5e is not convincing at all; how many mice have been electroporated? Some barrels seem to contain very little RORBi GFP and yet have different sizes compared to controls. This should also be illustrated in coronal sections with careful quantifications"

Response: We agree with the reviewer that further clarifications should be provided regarding this point. Our results inducing a *Rorb* gain of function both *in vitro* in dissociated thalamic cultures and in electroporated animals *in vivo* strongly suggest that the anatomical substrate for the increase in size of S1 is an augmented branching of thalamocortical axonal arbors. By performing *in utero* electroporation *in vivo*, the quantity of plasmid that each neuron receives is random and cannot be controlled. Therefore, the example image shown in Figure 5e cannot be used to determine the levels of GFP per neuron and barrel, as the entire axonal arbor is not covered in a single slice. Nevertheless, we have strengthened this point by performing single neuron axonal tracing and measured the extension of single VPM axons at the S1 barrel-field after *Rorb* gain of function *in vivo* (new data included in **Figure 6h** and **6i**). This new data strongly supports our previous *in vitro* and *in vivo* findings showing that *Rorb* functions to modify TCA axon complexity.

Point 12. "How do the authors reconcile the prenatal occurrence of the waves with postnatal changes in gene expression? Previous work has shown that S1 size is affected by enucleation postnatally, how do the current findings relate to this? What effect does a late postnatal enucleation have on S1 size? Exclusion of postnatal input-dependent processes is critical in this sense, and postnatal trimming is not sufficient to exclude this possibility (skin receptors remain activatable by direct contact). Caution of whisker follicles would be necessary to exclude this possibility. This should, at least, be discussed"

Response: In Fetter-Pruneda et al 2012, the authors found that enucleations after P7 in rat do not lead to expansion of the S1 cortical area size. Thus, at least in rats, enucleations after the first postnatal week do not engage a cortical effect. The stage during the first postnatal week up to which enucleations could still trigger a S1 cortical size change, needs to be determined but is beyond the scope of the current study. The phenomenon of the thalamic waves here described in mouse starts from E14.5 and lasts until P2. The changes in *Rorb* expression are seen from E16.5 up to P4. Thus, the observed changes in gene expression are fully overlapping with the time window of existence of the thalamic waves. It is tempting to speculate that after thalamic waves have ceased, sensory input depletion could not induce cross-modal adaptation of cortical area size, although this hypothesis will need to be investigated and is beyond the scope of the current study. Indirectly supporting this idea, ION transection in fetal, but not newborn, rats resulted in the absence of the vibrissae representation in S1 but did not result in a significant reduction of the total

S1 size (Killackey et al., 1994). Similarly, cauterization of whisker follicles generates an absence of barrels but not a change in size of S1 (Belford & Killackey, 1980; Bates & Killackey, 1987; Erzurumlu and Gaspar, 2012). Thus, we feel the proposed experiment would not lead to further conclusions.

Point 13. I could not find the Legends for Supplementary Figures.

Response: We apologize for the lack of clarity regarding this point. The Legends for Supplementary Figures are within the Supplementary info document provided.

REVIEWERS' COMMENTS:

Reviewer #1 (Remarks to the Author):

The authors have addressed all my comments and concerns. I congratulate the authors on this very nice study!

Reviewer #2 (Remarks to the Author):

The authors have improved the manuscript answered most of the concerns except adding in vivo data that would completely validate their models that calcium waves in the thalamus regulate cortical area size prior to sensory processing. However I understand that the in vivo experiments will take time and even without this data the conclusions are original and of interest to neuroscientists that study how brain wiring develops.

Reviewer #3 (Remarks to the Author):

The authors did a great job at revising the manuscript and I am satisfied with the current work. Please carefully check the manuscript for numerous typos and occasional awkward sentencings.

Point-by-point Response to Reviewers' Comments (NCOMMS-16-16239-T)

Reviewer #1

"The authors have addressed all my comments and concerns. I congratulate the authors on this very nice study!"

Response: We thank the reviewer for this positive comment.

Reviewer #2

"The authors have improved the manuscript answered most of the concerns except adding in vivo data that would completely validate their models that calcium waves in the thalamus regulate cortical area size prior to sensory processing. However I understand that the in vivo experiments will take time and even without this data the conclusions are original and of interest to neuroscientists that study how brain wiring develops."

Response: We thank the reviewer for this constructive comment.

Reviewer #3

"The authors did a great job at revising the manuscript and I am satisfied with the current work. Please carefully check the manuscript for numerous typos and occasional awkward sentencing."

Response: We thank the reviewer for this positive comment. We have revised the manuscript accordingly.